# Narrowing the Focus:
# Learned Optimizers for Pretrained Models

**Gus Kristiansen**[*]
Google DeepMind

**Mark Sandler**
Google DeepMind

**Andrey Zhmoginov**
Google DeepMind

**Nolan Miller**
Google DeepMind

**Anirudh Goyal**
Mila – Quebec AI Institute

**Jihwan Lee**
Google DeepMind

**Max Vladymyrov**[*]
Google DeepMind

## Abstract

In modern deep learning, the models are learned by applying gradient updates using an optimizer, which transforms the updates based on various statistics. Optimizers are often hand-designed and tuning their hyperparameters is a big part of the training process. Learned optimizers have shown some initial promise, but are generally unsuccessful as a general optimization mechanism applicable to every problem. In this work we explore a different direction: instead of learning general optimizers, we instead specialize them to a specific training environment. We propose a novel optimizer technique that learns a layer-specific linear combination of update directions provided by a set of base optimizers, effectively adapting its strategy to the specific model and dataset. When evaluated on image classification tasks, this specialized optimizer significantly outperforms both traditional off-the-shelf methods such as Adam, as well as existing general learned optimizers. Moreover, it demonstrates robust generalization with respect to model initialization, evaluating on unseen datasets, and training durations beyond its meta-training horizon.

## 1 Introduction

The optimization loop forms the backbone of machine learning algorithms. The choice of optimizer and its hyperparameters heavily influences the final model's performance. Despite its importance, optimizer selection often relies on heuristics and domain-specific knowledge. For instance, Adam optimizer (Kingma & Ba, 2014) excels with language problems, but struggles with image classification problems. The potential of second-order methods for stochastic data remains largely untapped, except for a few notable exceptions (Martens & Grosse, 2015; Gupta et al., 2018). Every year, more and more optimization methods are proposed, making it increasingly challenging for practitioner to select the best one (Choi et al., 2019; Schmidt et al., 2021).

Learned optimizers or learning-to-learn methods (Schmidhuber, 1987; Bengio et al., 1990; 2013) offer a potential solution to this challenge by automating the optimizer selection process. These methods automate this process by either learning the hyperparameters of existing optimizers or developing entirely new optimization algorithms. However, two significant challenges limit widespread adoption of learned optimizers: meta-optimization difficulties, and meta-generalization.

Meta-optimization difficulties arise from the sensitivity of gradients in bi-level optimization with a high number of training iterations (aka optimization horizon) (Metz et al., 2021). Evolutionary methods can help mitigate this because they do not rely directly on gradients, and thus are less susceptible to gradient sensitivity issues. However, the complexity of the problem increases with the length of the optimization horizon. Short horizon bias (Wu et al., 2018) constitutes another problem, where meta-optimization over a specified time horizon introduces a bias that prevents it to be applicable to a longer time horizon.

Meta-generalization refers to a learned optimizer's ability to perform well on novel tasks it was not specifically trained for. One approach to achieving this is to train the optimizer on the widest

---

[*]Correspondence to `gusatb@google.com`, `mxv@google.com`

possible range of tasks. Previous research focused on developing a single, adaptable algorithm by training it on a very broad domain. For example, Versatile Learned Optimizers (VELO, Metz et al., 2022) was trained on hundreds of different problems, ranging from simple linear regression to reinforcement learning. However, this "one-size-fits-all" approach has significant drawbacks. The computational cost of training such an optimizer is immense, requiring $4\,000$ TPU-months in the case of VELO. Furthermore, a single optimizer struggles to effectively adapt to the vast array of loss surfaces encountered across such a diverse set of tasks. This makes it challenging to achieve consistently high performance across all tasks.

Instead of aiming for broad applicability, we focus on the increasingly relevant domain of fine-tuning pretrained models. This specialization allows us to develop learned optimizers in a more targeted way. By narrowing the scope of the meta-training domain, the learned optimizer can specialize and excel in the specific types of tasks it is designed for. Fine-tuning also simplifies the optimization problem by leveraging existing knowledge encoded in pretrained checkpoints (Wei et al., 2021) and is typically done for a relatively few iterations, since only a small alignment with the current task is needed to achieve good results.

To achieve this, we introduce L3RS (Learned Layer-wise Learning Rate Scheduler, pronounced "lers"), a novel learned optimizer designed to leverage the performance of a set of base optimizers within a narrower domain. L3RS uses a variety of performance features, such as adaptive exponential moving averages (EMA) across different time scales. EMAs, or model averaging, have demonstrated improved generalization in real-world applications (Tarvainen & Valpola, 2017; Izmailov et al., 2018). Based on these features, L3RS produces layer-wise updates as a linear combination of the predefined base optimizers, along with a corresponding learning rate for each layer.

Our contributions are as follows.

- We propose a novel learning-to-learn method that leverages the strengths of multiple base optimization methods. This optimizer is simple and intuitive, having only a fraction of parameters compared to other black-box learned optimizers.

- We demonstrate that meta-training a learned optimizer on a narrow domain results in an optimizer that can outperform existing methods, with more that $50\%$ speedup compared to another learned optimizer baseline, and more than $200\%$ speedup compared to best performing traditional optimizers.

- We evaluate the meta-generalization of the proposed optimizer across several components: generalization to longer training horizon, different pretraining and model initializations, and a different evaluation dataset.

## 2  PRELIMINARIES

Consider a loss function $L(\theta_n, X)$, parameterized by weights $\theta_n$ for a given data $X$ at cetain optimization step $n$. Learned optimizers $f_\psi$, parameterized by $\psi$, provide a direction to update the model's weights $\theta_n$ as

$$\theta_{n+1} = \theta_n + f_\psi(\Phi; \theta_n, X), \tag{1}$$

where $\Phi$ is a collection of features that describes the current (or past) optimization statistics, such as the gradient or the loss value. For example, the simplest learned optimizer might simply optimize the learning rate $\lambda$ of SGD, as $f_\psi(\Phi; \theta_n, X) = \lambda \nabla L(\theta_n, X)$, in which case $\Phi := \{\nabla L\}$ and $\psi := \{\lambda\}$. In case of ADAM, the optimization parameters would be $\psi := \{\lambda, \beta_1, \beta_2\}$, where $\lambda$ is the learning rate, $\beta_1$ and $\beta_2$ are the exponential decay rates for the first and second moments.

In Figure 1 we show both the inner and the outer loop of a typical meta-training process. We train $\psi$ on a distribution of tasks from a given dataset. A task is a set $T := \{\theta_0, \{D_K^t\}, D^e\}$, where $\theta_0$ represents initial model weights (or a distribution of initial model weights), $\{D_K^t\}$ is a sequence of $K$ training batches, and $D^e$ represents an evaluation batch. A batch typically consists of data $X$ and, for classification problems, may also include labels. We assume that $\theta_0$ is given as input, and we have no control over its generation process (unlike initialization-based meta-learning approaches, such as MAML, Finn et al., 2017).

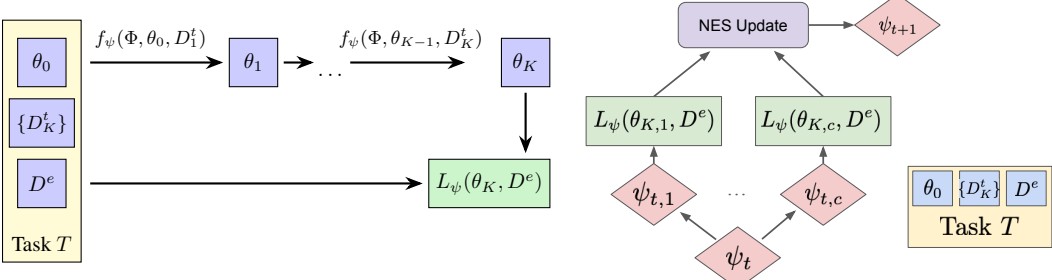

Figure 1: *Left:* Inner loop evaluation. Given a task $T := \{\theta_0, \{D_K^t\}, D^e\}$, the learned optimizer $f_\psi$ uses optimization statistics $\Phi$ to update model parameters starting from $\theta_0$ for each of $\{D_K^t\}$ batches. After $K$ update steps, the final model parameters are evaluated on the evaluation set $D^e$ using meta-loss $L_\psi(\theta_K, D^e)$. *Right:* Outer loop NES meta-training iteration. Given a task $T$ and meta-parameters $\psi_T$, Gaussian noise is added to $\psi_t$ to produce a number of candidates equal to the population size $c$, $(\psi_{t,0}, ..., \psi_{t,c})$. An inner loop evaluation is performed on the given task for all candidates. The fitness of each candidate is then used to perform an NES update step, resulting in the next learned optimizer parameters, $\psi_{t+1}$.

Learned optimizer parameters $\psi$ are evaluated by performing $K$ update steps (equation 1) on every batch from $D_K^t$, to get $\theta_K$ and calculating the loss using the evaluation batch $L_\psi(\theta_K, D^e)$.

We use Natural Evolution Strategies (NES, Salimans et al., 2017) to meta-train the learned optimizers. We define a task meta-loss $L_\psi^M(T) = L_\psi(\theta_K, D^e)$, the loss on the evaluation batch after K training steps of a task $T = \{\theta_0, \{D_K^t\}, D^e\}$. We then define a fitness function $F(\psi, (T_1, ..., T_b)) = -\frac{1}{b}\sum_{i=1}^{b} L_\psi^M(T_i)$, using a batch of $b$ tasks for each generation. NES is used to maximize the fitness function during meta-training.

After the learned optimizer is trained, it can be evaluated on a new task distribution $\widetilde{T} := \{\widetilde{\theta}_0, \{\widetilde{D}_K^t\}, \widetilde{D}^e\}$.

## 3 MOTIVATION AND RELATED WORK

We can separate the optimizers into two main categories. First category, so called black-box optimizers (Li & Malik, 2016; Andrychowicz et al., 2016; Wichrowska et al., 2017; Lv et al., 2017; Sandler et al., 2021; Metz et al., 2020a;b; 2022), learns an update function $f_\psi$ from scratch using a custom inner optimization loop. These methods often have a large number of parameters, making them prone to overfitting and stability issues (Harrison et al., 2022). While recent work has explored using Transformers as a meta-learner architecture (Chen et al., 2022; Moudgil et al., 2023; Jain et al., 2024), they have not yet shown a significant advantage over traditional optimizers or other learned optimizer architectures.

A notable example of black-box optimizer is VELO, which uses per-tensor HyperNetworks (Ha et al., 2016) represented by 512-wide LSTMs to generate parameters of per-parameter multi-layer perceptrons (MLPs). To compute the parameter updates, it passes the per-parameter features through the generated MLP network consisting of 2-hidden layer, 4-node MLP. The input to the HyperNetwork and MLP vary slightly, but generally represent the training dynamic of the optimization. VELO is designed to be general and has $\sim 2.3$ million parameters that need to be learned, which makes meta-training very expensive. We envision an ideal optimizer would have a less complicated design, fewer parameters and, ideally, would be fast to train on a narrow task domain.

Other techniques utilize algorithm discovery as a program search (Bello et al., 2017; Wang et al., 2022; Zheng et al., 2022; Chen et al., 2023), where the symbolic optimizers are found using a tree search of predefined operations (such as gradient and momentum).

The second category of meta-optimizers learns a higher-level meta-component on top of existing hand-designed optimizers. Such approaches include learning to adapt the hyperparameters of an

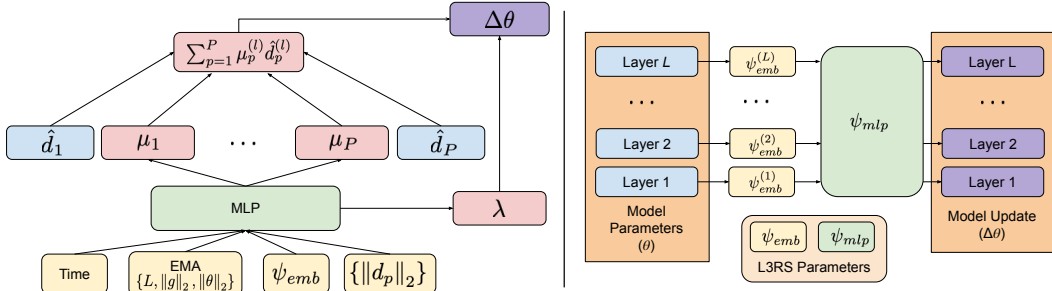

Figure 2: *Left:* L3RS applied to a single layer of the target model. The MLP receives time features, EMA features of the global loss, layer gradient norm and layer parameter norm, the target layer embedding, and the norm of each direction provided. The MLP outputs the weighting for each direction ($\mu_p$) as well as the final update norm ($\lambda$). *Right:* L3RS is applied at every layer of the network independently with the same MLP weights but different layer embeddings.

underlying optimizer Daniel et al. (2016), or learn a schedule for the learning rate (Xu et al., 2017; 2019). In addition, these methods can include adapting the optimizer's parameters directly (Shaban et al., 2019). Hypergradient methods (Maclaurin et al., 2015; Baydin et al., 2017; Grazzi et al., 2020; Moskovitz et al., 2019) perform a hyperparameter search on-the-fly during optimization.

Our proposed approach strikes a balance between flexibility of black-box minimizer and the stability of learning hyperparameters of known optimizers. We leverage the benefits of hand-designed optimizers while incorporating the adaptability of learned components using a simple small neural network for learning the per-layer learning rate. Unlike existing methods that learn a fixed set of hyperparameters for a single existing optimizer, L3RS learns to utilize a set of base-optimizers. Moreover, during meta-training we also search over the hyperparameters of the base-optimizers.

Almeida et al. (2021) uses an LSTM controller to adjust the hyperparameters of a hand-designed inner-optimizer, which uses a collection of hand-designed parts from different known optimizers. A major difference with our method is that L3RS operates per-layer and that our algorithm wraps existing optimizers out-of-the box, without the need to manually add features.

Prémont-Schwarz et al. (2022) also propose an in-between approach, where the learned optimizer falls back to the base-optimizer. However, their goal was to ensure convergence, not necessarily to improve the performance.

The L3RS architecture was also partially motivated by Learning Rate Grafting (Agarwal et al., 2022), which demonstrates the utility of isolating the direction of an optimizer from its magnitude.

Similar to our paper, Landro et al. (2021) also propose an algorithm that combines Adam and SGD as a linear combination of their update direction. However, their method differs significantly in two key aspects: (1) they do not provide a mechanism for learning the mixing coefficient, instead treating it as fixed hyperparameters, and (2) their combination is applied globally across all layers, while our proposed method employs a layer-wise adaptive strategy.

## 4 L3RS: LEARNED LAYER-WISE LEARNING RATE SCHEDULER

### 4.1 MODEL ARCHITECTURE

Given a set of update vectors $d_p$ from $P$ input base optimizers (e.g. SGD or Adam) L3RS computes a model parameter update step per layer $l$ using:

$$\Delta\theta^{(l)} = \lambda^{(l)} \sum_{p=1}^{P} \mu_p^{(l)} \hat{d}_p^{(l)}, \tag{2}$$

where $\lambda^{(l)}$ is a layer-specific update $l_2$-norm, $\mu_p^{(l)}$ are normalized mixing coefficients for the base optimizers' directions $\sum_{p=1}^{P} \mu_p^{(l)} = 1$, and $\hat{d}_p^{(l)}$ are normalized optimizers' direction $\hat{d}_p^{(l)} = \frac{d_p^{(l)}}{\left\| d_p^{(l)} \right\|_2}$.

Figure 2 shows the main input and output components of L3RS as well as provides a flow-chart on how the weights of the underlying model $\theta$ are updated.

In order to learn $\lambda^{(l)}$ and $\mu_p^{(l)}$, we provide L3RS with a feature vector, comprising the adaptive EMA, time, embedding features as well as the magnitude of the update directions defined below. This input is then processed by two fully connected layers with sizes 32 and 16, using ReLU activation. The model outputs logits $z^{(l)} \in \mathbb{R}^{P+1}$, from which we reconstruct $\mu_p^{(l)} = \frac{\exp(z_p^{(l)})}{\sum_{j=1}^{P} \exp(z_j^{(l)})}$ and $\lambda^{(l)} = \exp(z_{P+1}^{(l)})$. The hyperparameters of the input optimizer (e.g. Adam's $\beta_1$ and $\beta_2$)) are also jointly meta-learned along with the L3RS's model parameters. This enables co-adaptation of the layer-wise scaling and the underlying optimization algorithm.

**Adaptive Exponential Moving Averages (EMA).**  In order to capture both short-term and log-term performance trends, for each layer $l$, we maintain three EMAs with different smoothing factors $\gamma \in \{0, 0.9, 0.99\}$. EMAs are updated recursively as $a_{i,k+1}^{(l)} := \gamma_i a_{i,k}^{(l)} + (1 - \gamma_i)\xi^{(l)}$, where $a_{i,k}^{(l)}$ denotes the $i$th EMA for layer $l$ at time step $k$, and $\xi^{(l)}$ represents a layer-specific statistic of interest. Specifically, for a given layer we track the following:

- *Log $l_2$-norm of weights.* $\log(\|w^{(l)}\|_2)$, where $w^{(l)}$ are the weights for layer $l$.
- *Log $l_2$-norm of gradients.* $\log(\|g^{(l)}\|_2)$, where $g^{(l)}$ are the gradients for layer $l$.
- *Loss.* The loss of the overall model (shared across all layers).

To ensure unbiased estimates, especially during the initial time steps, we apply a bias correction to the EMAs before incorporating them as input features: $\tilde{a}_k^{(l)} = \frac{a_k^{(l)}}{1 - \gamma^k}$.

**Time features.**  Inspired by VELO (Metz et al., 2022), we propose incorporating the following time features for the explicit modeling of temporal dynamics of the optimization progress. These features enable the optimizer to adapt its strategy based on the stage of the training process. These features are generated using $k$, the current training step, and $K$, the number of total training steps. Similar to some learning rate schedulers, $K$ must be given to the optimizer at the beginning of training so that these time features can be generated.

- *Relative time features.* $\tanh\left(10\left(\frac{k}{K} - \alpha_i\right)\right)$, where $\alpha_i$ are 11 linearly scaled reference points between $0.0$ and $1.0$, inclusive. These features capture the relative progress through a sequence, with the hyperbolic tangent function providing a smooth, bounded representation. Figure 7 in the Appendix shows the values of these features throughout training.
- *Absolute time features.* $\tanh\left(\log\left(K\beta_i\right)\right)$, where $\beta_j$ are 4 log-scaled scaling factors between $0.0001$ and $0.1$. These features provide a logarithmic encoding of the total duration, which can be useful for appropriate learning rate scaling. For longer time horizons these features can be extended such that $\max(\beta) > \max(K)$. Figure 8 in the Appendix shows the values of these features for various total training lengths $K$.

**Embedding features.**  A 16-dimensional embedding vector $\psi_{emb}$ is meta-learned for each layer, allowing the optimizer to learn specialized per-layer dynamics.

**Base-optimizer direction magnitude.**  For each base optimizer, we provide the log $l_2$-norm of its update: $\log(\|d_p\|_2)$.

## 4.2 Comparison with VELO

VELO was designed with an efficient hypernetwork-style architecture. The majority of the computation cost is reduced to per-layer LSTM networks, which then generate cheaper per-parameter networks which output the update step direction. L3RS is able to leverage smaller MLP networks for the per-layer operations, and can rely on base optimizers rather than generating the update step direction manually. These design choices result in a learned optimizer which will typically have two to three orders to magnitude fewer parameters compared to VELO.

| Optimizer | Memory Overhead | Compute Overhead |
|-----------|-----------------|------------------|
| SGD | $0.0\times$ | $0.003\%$ |
| ADAM | $2.0\times$ | $0.019\%$ |
| L3RS | $2.0\times$ | $0.057\%$ |
| VELO | $4.0\times$ | $0.920\%$ |

Table 1: Memory and compute overhead for various optimizers using a ResNet-34 model with 25 output classes. Memory Overhead is the ratio of optimizer state size to model parameter size. Compute overhead is the compute cost ratio of the optimizer update step to a full training step for a batch of 64 images. Compute cost is calculated using Jax compilation statistics.

## 5 EXPERIMENTS

To evaluate the performance of our proposed L3RS optimizer, we designed a series of fine-tuning image classification experiments using the ResNet-34 model.

**Fine-Tuning.** Our experiments focus on fine-tuning pretrained models. In particular, we are interested in optimization horizons on the order of hundreds or thousands of steps. While training can certainly extend further, this training regime is relevant in cases where the amount of training data or high computational costs can be a limiting factor.

**Model Choices.** While our proposed L3RS can wrap any number of base-optimizers, we found that even a simple version that combines together only ADAM and SGD (without momentum) directions performs well. For the remainder of this paper, when we refer to L3RS, we specifically mean this ADAM and SGD combination unless we specify otherwise.

During inference, the memory footprint of this L3RS optimizer state is $2\times$ the model parameters (all from ADAM). VELO uses a memory state of $4\times$ the model parameters (plus additional memory from 3 AdaFactor style accumulators). Table 1 shows a comparison of the memory state requirements for various optimizers, as well as the compute overhead for each.

We derive updates separately for every convolutional, dense or batch normalization layer of the model, including separate updates for kernels and biases. For ResNet-34, this results in $L = 111$ components for which we compute learning rate $\lambda^{(l)}$ and mixing coefficients $\mu_p^{(l)}$, for $l = 1, \ldots, L$. With $P = 2$ optimizers, L3RS outputs 333 parameters per iteration.

For NES meta-training, we use exponential decay for both meta-learning rate $\alpha$ and the Gaussian noise standard deviation $\sigma$ by 0.5 every 500 generations. We use the same antithetic sampling and fitness transformations as in Salimans et al., 2017. We use a population size of 32, meta-batch size of 4 and train for 2000 generations for all our experiments (except the ablation study). We parallelize the evaluation of each candidate in the population, using a total of 32 A100 GPUs for four days.

**Task Distribution and Meta-Training.** We trained the learned optimizer using IMAGENET dataset in the following way. We partition the IMAGENET dataset into three distinct subsets: pretraining, meta-train (IMAGENET25), and meta-test (IMAGENET25EVAL). The first 500 classes are used to pretrain a model using a conventional off-the-shelf algorithm. The pretrained model serves as an initialization to our method. The remaining 500 classes are randomly divided into meta-train and meta-test sets. Details on checkpoint pretraining are provided in Appendix A.

A task within our meta-learning framework is constructed by randomly sampling 25 classes from either the IMAGENET25 or IMAGENET25EVAL set. The selected classes are used to generate $K$ training batches with 64 samples each and 1 evaluation batch with 256 samples, using the train and validation splits respectively. During meta-training, the number of training batches $K$ is uniformly sampled from a range of 10 to 500, simulating diverse fine-tuning scenarios.

We meta-train the L3RS optimizer and the VELO optimizer on the IMAGENET25 meta-train task distribution using NES as described above.

**Meta-Evaluation and Baselines.** To evaluate the performance, we check for in-distribution and out-of-distribution generalization. For *in-distribution evaluation* we use the same initialization as the meta-

training ($\widetilde{\theta}_0 = \theta_0$) and use IMAGENET25EVAL tasks for meta-test. For *out-of-distribution evaluation* we use the PLACES (López-Cifuentes et al., 2020) dataset. We create a separate PLACES initialization by pretraining ResNet-34 on the first 150 classes of PLACES. We use the remaining classes as a meta-test set (PLACES25EVAL). Importantly, this dataset is entirely novel and unseen during the meta-training, which allows us to evaluate out-of-distribution performance of the optimizers.

We compare L3RS against the following baselines:

- VELO (ft). An instance of VELO meta-trained using the same method as L3RS.
- VELO (og). The original pretrained VELO model (Metz et al., 2022), representing a general-purpose learned optimizer.
- VELO (og, Head). The original pretrained VELO model applied to only the final model layer (freezing the rest of the model).
- ADAM (cosine). Adam optimizer with a cosine learning rate schedule.
- ADAM (cosine, Head). Adam optimizer with a cosine learning rate schedule applied to only the final model layer (freezing the rest of the model).
- ADAM (const). Adam optimizer with constant learning rate.
- ADAM (const, Head). Adam optimizer with constant learning rate appled to only the final model layer (freezing the rest of the model).

For optimizers which depend on the number of training steps $K$, such as ADAM (cosine), VELO (ft) and L3RS, we perform separate evaluations across various $K$ values to assess their performance both within and beyond the meta-training regime.

For all meta-evaluations we average the results of 100 tasks sampled from the specified task distribution. The average and standard deviation of eval accuracy across the sampled tasks is reported. All figures are reproduced using loss instead of accuracy in Appendix A.

**Results.** Figure 3 shows the main result. Our goal is to assess the performance of the learned optimizer under various conditions, for both in and out of distribution scenarios. (**A**) Represents in-distribution evaluation, where the initialization, evaluation dataset, and the number of steps match the meta-training regime. We also explore out-of-distribution performance by extending the number of evaluation steps beyond the meta-training range (indicated by the dashed line in all plots), changing the evaluation dataset to PLACES25EVAL (**B**), modifying the initialization to utilize a model pretrained on PLACES (**C**), or using both the PLACES checkpoint and PLACES25EVAL dataset (**D**).

First, comparing VELO (ft) to VELO (og), we observe that fine-tuning VELO to a specific dataset and number of steps leads to a significant performance improvement compared to the general-purpose VELO (og). However, VELO (ft) performance deteriorates sharply when evaluated outside its meta-training regime. We hypothesize this is due to VELO complexity and parameter count, making it more sensitive to deviations from its training distribution, especially number of steps.

Despite the improvement from fine-tuning, VELO (ft) still underperforms compared to L3RS. We attribute this to L3RS's more lightweight architecture and its inherent ability to fallback to the performance of its base optimizers for robust performance. L3RS also performs favorably against ADAM for all $K$ values within the training distribution (10 to 500 steps), but their performance becomes comparable when evaluated beyond $K = 1\,000$ steps.

Interestingly, changing the initialization or evaluation dataset to PLACES has a minimal effect on the relative performance of the optimizers, even though PLACES was not included in the meta-training data. This highlights the robust generalization capabilities of both L3RS and the fine-tuned VELO.

We also apply the methods to a randomly initialized Resnet-34 model (**E**). RandomInit uses a standard initialization method (Klambauer et al., 2017). In this case, L3RS does not outperform the baseline methods. We believe this is because, while L3RS generalizes well to different checkpoints (**C**), the randomly initialized model may be far out of distribution for L3RS to perform optimally.

In Figure 3 (**F**) we show the speedup, in training steps, L3RS achieves over baseline methods to reach a given accuracy. L3RS is able to demonstrate robust 50% speedup over VELO (ft) and $100\%$ speedup over the best hand-designed optimizer.

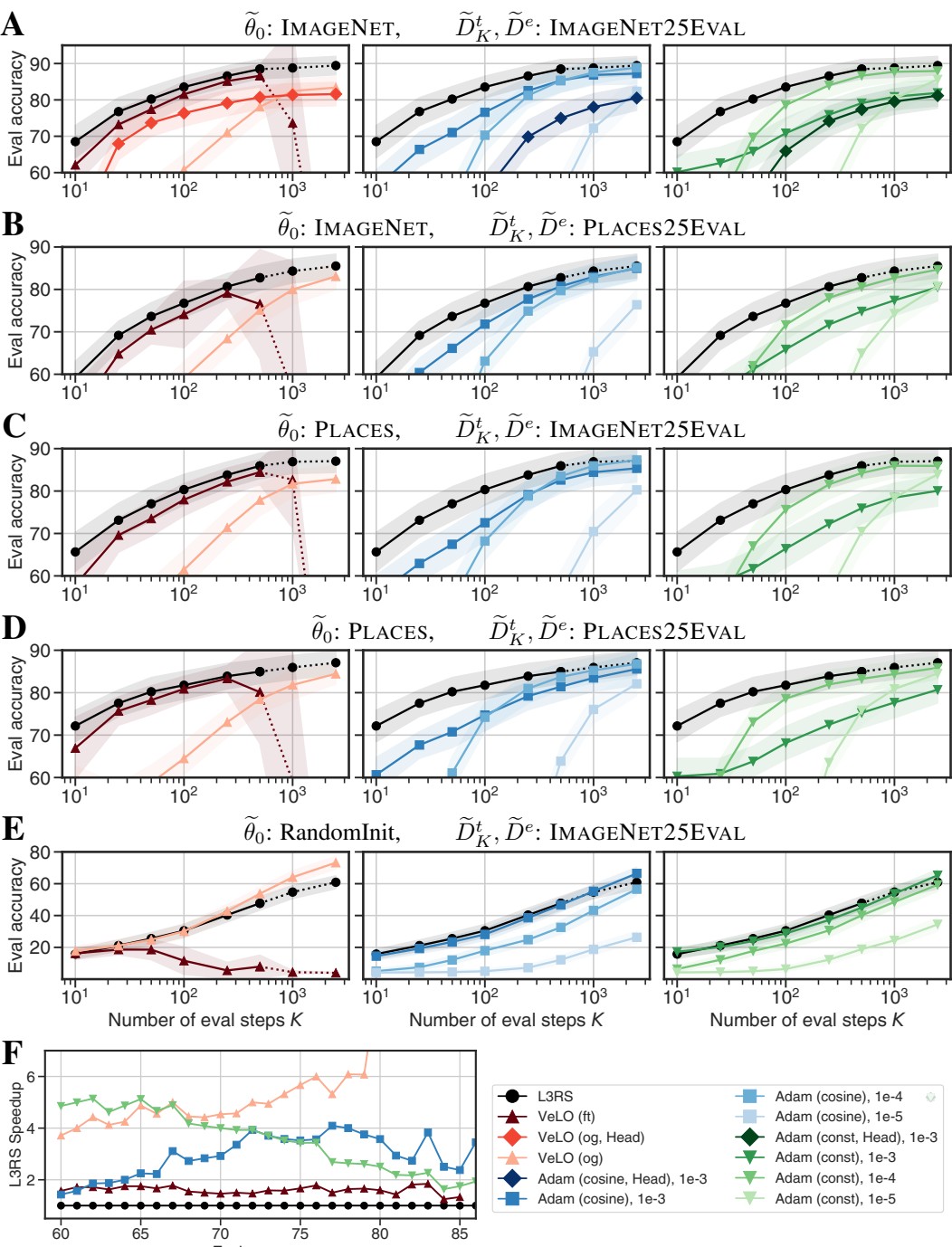

Figure 3: Meta-evaluation of L3RS meta-trained on IMAGENET25 for 10 to 500 steps along with various benchmarks. Performance is compared to VELO (*left*), ADAM with cosine learning rate (*center*), and ADAM with a constant learning rate (*right*). Each marker represents model evaluation at that number of steps. Solid lines indicate the number of steps for in-distribution evaluation, while dashed lines indicate generalization to more steps than meta-training.

**A. In-domain Generalization.** Both initialization and evaluation are on IMAGENET.

**B. Out-of-Domain Initialization.** Initialized on IMAGENET, evaluated on PLACES25EVAL.

**C. Out-of-Domain Evaluation.** Initialized on PLACES, evaluated on IMAGENET25EVAL.

**D. Out-of-Domain Init & Eval.** Both initialization and evaluation are on PLACES dataset.

**E. Random Initialization.** Random initialization, evaluated on IMAGENET25EVAL dataset.

**F. Speedup of L3RS in iterations.** For in-domain generalization, this shows how much faster L3RS achieves a given accuracy compared to the baselines.

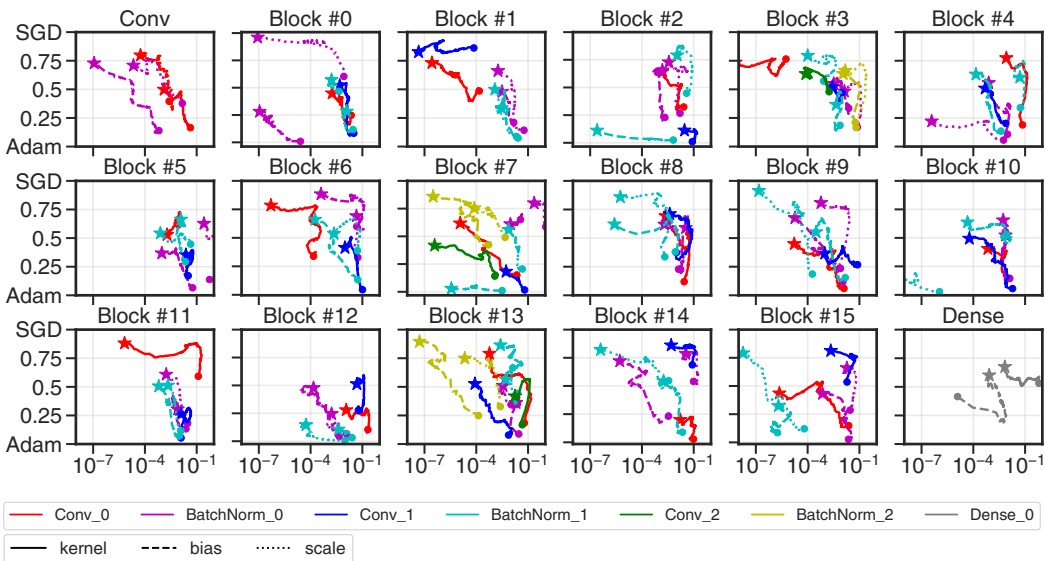

Figure 4: Visualization of learned mixing coefficients $\mu^{(l)}$ and per-layer learning rates $\lambda^{(l)}$ over 100 steps for a ResNet-34 model. Each layer's type and component are distinguished by color and line type (see legend). The general trend shows curves moving up and to the left, indicating a transition from ADAM ($\mu^{(l)} = 0$) to SGD ($\mu^{(l)} = 1$) and a decrease in $\lambda^{(l)}$. The initial step is marked with a $\bullet$ and the final step with a $\star$.

| Base-optimizers | Embedding | No Embedding | Per-layer MLP | Global |
|---|---|---|---|---|
| SGD only | $68.18 \pm 4.29$ | $64.48 \pm 5.05$ | $64.71 \pm 5.00$ | $64.37 \pm 4.94$ |
| Adam only | $68.52 \pm 4.25$ | $61.44 \pm 5.14$ | $67.14 \pm 4.57$ | $63.45 \pm 4.84$ |
| SGD, Adam | $\mathbf{68.93 \pm 4.58}$ | $65.70 \pm 4.91$ | $68.52 \pm 4.40$ | $66.58 \pm 4.93$ |

Table 2: Average and standard deviation of evaluation accuracy for different optimizers and per-layer strategies.

The learned parameters of L3RS exhibit interesting dynamics during a 100-step evaluation (Figure 4). Initially, the optimizer strongly favors the ADAM direction with a high learning rate for most layers. As training progresses, this preference shifts, transitioning towards SGD and a lower learning rate (represented by a movement towards the top-left of the plot). Figure 5, showing the average parameter movement across all layers, reveals several distinct phases: an initial warm-up period with an increasing learning rate, a period of relatively constant learning rate while transitioning from ADAM to SGD, a phase of rapid learning rate decay, and a final convergence to the SGD direction over the last $\sim$10 steps. While the precise interpretation of these parameter dynamics is challenging, the L3RS parameters are considerably more interpretable than those of most black-box learned optimizers.

For additional experiments using ResNet-18 and Vision Transformer (ViT) (Dosovitskiy, 2020) architectures, please refer to Appendix E. Adabelief (Zhuang et al., 2020) is also included as an additional baseline in the evaluation tables provided in Appendix D.

## 6 ABLATIONS AND VARIANTS

We perform a methodical ablation and variant study to justify the design choices of the L3RS architecture. We meta-train all models in this section using the IMAGENET25 task distribution but set number of steps, $K$, to 10. We additionally meta-train for 500 generations of NES and change the meta-learning rate decay rate steps from 500 to 100.

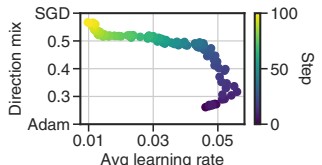

Figure 5: Average learning rate vs. direction mix between Adam and SGD for each of the 100 steps of the L3RS optimizer.

| EMA's Smoothing Factors | Accuracy |
|---|---|
| $\gamma \in \{0.99, 0.9, 0.0\}$ | $68.93 \pm 4.34$ |
| $\gamma \in \{0.9, 0.0\}$ | $67.43 \pm 4.64$ |
| $\gamma \in \{0.0\}$ | $67.78 \pm 4.47$ |
| $\gamma \in \varnothing$ | $66.52 \pm 4.84$ |

Table 3: Average and standard deviation evaluation accuracy given adaptive EMA features.

| Base Optimizer Set | Accuracy |
|---|---|
| SGD, ADAM | **$68.93 \pm 4.58$** |
| ADAM, LION, LAMB | $68.61 \pm 4.50$ |
| ADAMAX, SGD, LAMB | $66.95 \pm 4.80$ |
| SGD, ADAM, ADAMAX, LION, LAMB, WeightDecay | $68.41 \pm 4.65$ |

Table 4: Average eval accuracy and standard deviation of L3RS variants, each using a different set of base-optimizers.

**Ablation of Base Optimizers.** We will show the results of using either SGD or ADAM alone as the wrapped optimizer rather than both together.

**Embedding Variants.** Without layer embeddings, L3RS struggles to distinguish between layers, relying heavily on adaptive EMA features. Alternatively, we can meta-learn a separate MLP for each layer of the target model. This significantly increases the optimizer's parameter count. We compare this per-layer MLP approach with a single shared MLP (without layer embeddings). We additionally report results for a Global method, which uses a single learning rate ($\lambda$) and direction weights ($\mu_p$) for all layers. Results for these different optimizer configurations are presented in Table 2.

**Ablation of Adaptive EMA Input Features.** We also investigated the impact of different smoothing factors ($\gamma$) for the adaptive exponential moving average (EMA) features. The standard L3RS uses $\gamma \in \{0.99, 0.9, 0.0\}$. For comparison, we trained models with $\gamma \in \{0.9, 0.0\}$, $\gamma = 0.0$ (representing raw features without averaging), and no adaptive EMA features ($\gamma = \emptyset$). The results are summarized in Table 3.

**New Directions.** Our main results are based on L3RS which uses only SGD and Adam as given directions. There is a lot of room for exploration in which optimizers and combinations will result in the most powerful L3RS variant for a given task. As an example of the flexibility of the architecture we explore a few combinations here. We leverage the following optimizers: LION (Chen et al., 2023), LAMB (You et al., 2020), ADAMAX (Kingma & Ba, 2014), as well as a WeightDecay direction which simply provides the negative direction of the current parameters. We use the same training/evaluation set up as the rest of the ablations and variants. We report the results in Table 4.

## 7 CONCLUSIONS

Learned optimization is a powerful paradigm which has the potential to outperform existing methods. Our results suggest that narrowing the domain that learned optimizers are meta-trained on can provide a reduced meta-training cost, and allow learning domain specific exploitations that can increase their performance, especially on fine-tuning tasks. We provide results repurposing VELO, an architecture designed for general learned optimization, as well as propose a new architecture L3RS, designed to take advantage of the narrow domain. We demonstrate the robustness of this method on out of distribution tasks. As fine-tuning tasks continue to become more relevant, especially for LLMs, this paradigm and architecture provides a promising direction for future research. Improving this technique further can result in faster model training, better performance, and robust data generalization.

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

## A  CHECKPOINT PRETRAINING

We create the pretrain checkpoints for both IMAGENET and PLACES using the same hyperparameters. We train the models using batch size 128 and train for 100,000 steps. Augmentations/preprocessing used include random cropping to size 224, random mirror, resize, and normalization (based on Imagenet train statistics). For the IMAGENET checkpoint trained on the first 500 classes, we reach an evaluation loss of 0.9387 and an evaluation accuracy of 75.68. For the PLACES checkpoint trained on the first 150 classes, we reach an evaluation loss of 1.312 and an evaluation accuracy of 61.72.

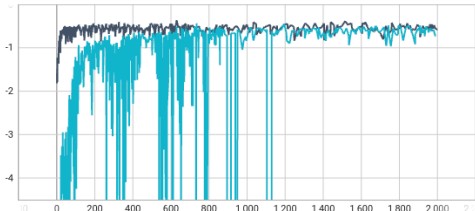 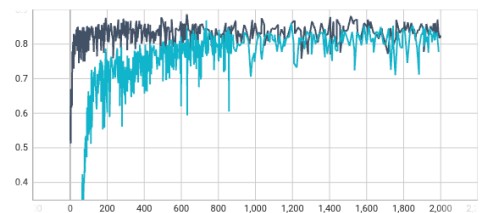

Figure 6: Meta-training curves for L3RS (Black), and VELO (Blue). Left: Average population fitness. Right: Average population evaluation accuracy.

## B    META-TRAINING

We provide the meta-training curves, both fitness and accuracy, for the two meta-trained learned optimizers. Figure 6 shows the curves for L3RS and VELO throughout training. VELO training is very unstable in this setup, though does converge by the end of training. L3RS reaches a high fitness very rapidly then slowly improves throughout the rest of the training, likely due to the smaller number of parameters in the learned optimizer.

Figures 7 and 8 show the example of the relative and absolute time features used by L3RS.

## C    LEARNED MIXING COEFFICIENTS DYNAMICS

To better understand the optimization dynamics of L3RS during a 100-evaluation run, we visualize the learning rate and direction mix per layer. Figure 9 provides a more granular view, displaying the same relationship for each individual layer.

## D    EVALUATION TABLES

We provide the full evaluation tables of accuracy and loss from the main evaluation experiments Figure 3 **A**-**E**. We additionally reproduce Figure 3 using the evaluation losses in Figure 10. Table 5 and Table  6 provide the accuracy and loss results respectively from the in-domain evaluation **A**. Table 7 and Table 8 provide the accuracy and loss results respectively for all head-only fine-tuning for evaluation **A**. Table 9 and Table 10 provide the accuracy and loss results respectively from the dataset-generalization evaluation **B**. Table 11 and Table 12 provide the accuracy and loss results respectively from the checkpoint-generalization evaluation **C**. Table 13 and Table 14 provide the accuracy and loss results respectively from the initialization and dataset-generalization evaluation **D**. Table 15 and Table 16 provide the accuracy and loss results respectively from the RandomInit evaluation **E**.

We have also provided results for Adabelief as an additional baseline for the main experiments. Adabelief tends to perform similar or slightly worse than Adam with Cosine learning rate decay.

## E    ADDITIONAL ARCHITECTURE EXPERIMENTS

We also provide experimental results for ResNet-18 and ViT models. The ViT architecture used is the S/16 model. The same meta-training hyperparameters are used, including the process for creating the pre-trained checkpoints. Unlike the main experiments, for evaluation 10 tasks are sampled rather than 100 due to resource and time constraints.  The ResNet-18 and ViT experiments show similar results to the main experiments. For all $K$ values within the training distribution (10 to 500 steps), L3RS performs favorably. As $K$ grows and exceeds 1000 steps, the performance becomes comparable. Similar to the evaluation tables listed above, we provide the accuracy and loss for these experiments. Table 17 and Table 18 provide the accuracy and loss results respectively for the ResNet-18 experiment. Table 19 and Table 20 provide the accuracy and loss results respectively for the ViT experiment.

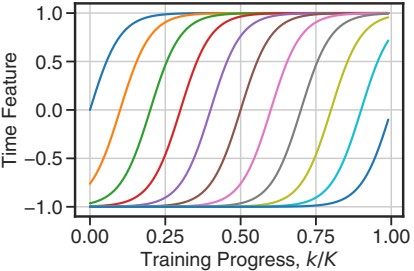

Figure 7: Relative time features as a function of training progress (k/K). Each of the lines represents a time input feature to the model, generated using $k$, $K$, and $\alpha_i$.

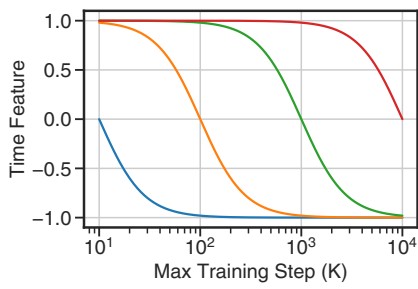

Figure 8: Absolute time features as a function of max training step (K). Each of the lines represents a time input feature to the model, generated using $K$ and $\beta_j$.

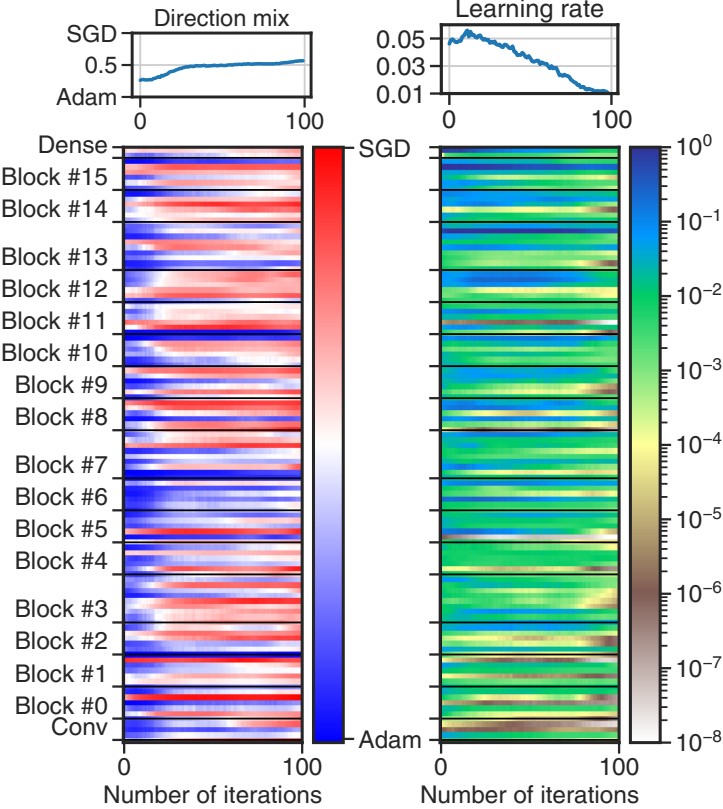

Figure 9: Learned mixing coefficients $\mu_p^{(l)}$ and $\lambda^{(l)}$ for each iteration of L3RS for ResNet-34 model. *Top:* average across all the layers. *Bottom:* individual for every learned component.

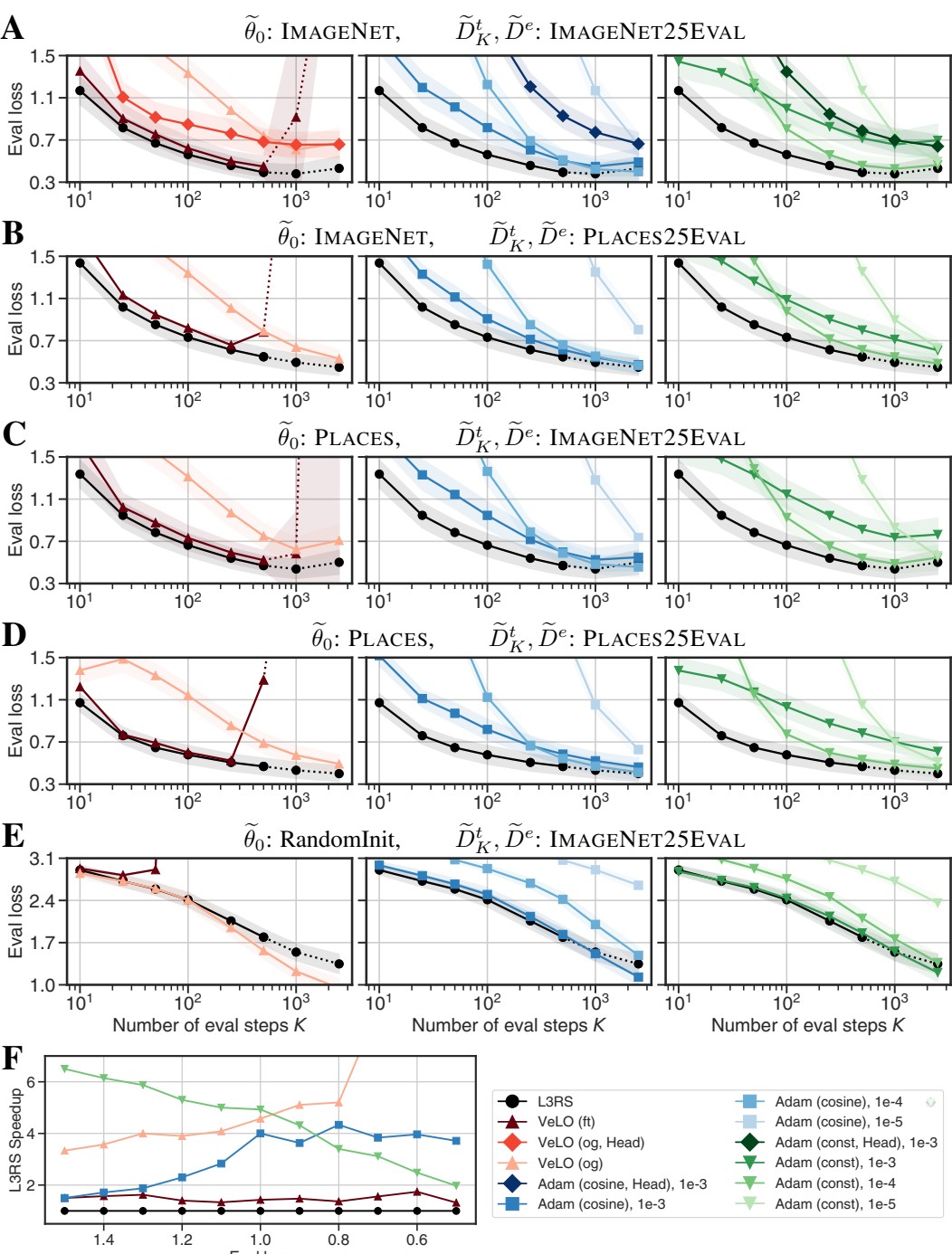

Figure 10: Meta-evaluation of the L3RS, meta-trained on IMAGENET25 for 10 to 500 steps along with various benchmarks. Performance is compared to VELO (*left*), ADAM with cosine learning rate (*center*), and ADAM with a constant learning rate (*right*). Each marker represents model loss evaluation at that number of steps. Solid lines indicate the number of steps for in-distribution evaluation, while dashed lines indicate generalization to more steps than meta-training.
**A. In-domain Generalization.** Both initialization and evaluation are on IMAGENET.
**B. Out-of-Domain Initialization.** Initialized on IMAGENET, evaluated on PLACES25EVAL.
**C. Out-of-Domain Evaluation.** Initialized on PLACES, evaluated on IMAGENET25EVAL.
**D. Out-of-Domain Init & Eval.** Both initialization and evaluation are on PLACES dataset.
**E. Random Initialization.** Random initialization, evaluated on IMAGENET25EVAL dataset.
**F. Speedup of L3RS.** For in-domain generalization, this shows how much faster L3RS achieves a given loss compared to the baselines.

| Optimizer | 10-Step | 25-Step | 50-Step | 100-Step | 250-Step | 500-Step | 1000-Step | 2500-Step |
|---|---|---|---|---|---|---|---|---|
| L3RS | 68.52± 4.46 | 76.78± 3.83 | 80.22± 3.45 | 83.54± 2.99 | 86.57± 2.77 | 88.47± 2.64 | 88.79± 2.57 | 89.44± 2.52 |
| VeLO (ft) | 62.11± 5.51 | 73.23± 4.29 | 77.41± 3.44 | 81.54± 3.46 | 85.16± 2.93 | 86.64± 3.1 | 73.64± 26.26 | 4.0 ± 0.17 |
| VeLO (og) | 50.63± 4.9 | 47.48± 5.2 | 53.7 ± 4.6 | 60.79± 4.24 | 71.02± 4.08 | 78.09± 3.62 | 82.45± 3.19 | 83.39± 3.04 |
| Adam (cosine), 1e-3 | 55.86± 4.92 | 66.38± 4.7 | 71.03± 4.11 | 76.59± 4.19 | 82.53± 3.46 | 85.34± 3.17 | 86.84± 2.7 | 87.25± 2.88 |
| Adam (cosine), 1e-4 | 10.69± 2.52 | 28.22± 4.03 | 52.11± 5.04 | 70.29± 4.6 | 81.22± 3.43 | 85.3 ± 3.22 | 87.52± 2.96 | 88.88± 2.59 |
| Adam (cosine), 1e-5 | 4.32 ± 1.45 | 5.09 ± 1.62 | 6.47 ± 1.81 | 10.66± 2.29 | 29.4 ± 4.0 | 54.18± 5.05 | 72.22± 4.67 | 82.36± 3.52 |
| Adam (const), 1e-3 | 60.12± 4.62 | 62.65± 4.37 | 65.82± 4.24 | 70.85± 4.23 | 75.92± 3.94 | 79.11± 3.7 | 80.87± 3.58 | 81.92± 3.61 |
| Adam (const), 1e-4 | 22.09± 3.54 | 51.72± 4.98 | 69.72± 4.59 | 78.58± 3.95 | 84.02± 3.07 | 86.65± 2.93 | 87.72± 2.76 | 87.95± 2.78 |
| Adam (const), 1e-5 | 4.83 ± 1.56 | 6.47 ± 1.77 | 10.7 ± 2.29 | 22.66± 3.62 | 53.91± 4.99 | 72.15± 4.6 | 80.34± 3.62 | 85.62± 3.16 |
| Adabelief, 1e-3 | 57.83± 4.65 | 59.53± 4.81 | 63.91± 4.72 | 69.93± 3.92 | 75.3 ± 3.98 | 78.57± 3.56 | 80.36± 3.42 | 81.73± 3.46 |
| Adabelief, 1e-4 | 26.67± 3.84 | 57.13± 4.93 | 71.93± 4.52 | 79.29± 3.89 | 84.3 ± 3.09 | 86.79± 2.85 | 87.8 ± 2.71 | 87.88± 2.69 |
| Adabelief, 1e-5 | 5.03 ± 1.59 | 7.19 ± 1.84 | 13.32± 2.73 | 29.51± 4.13 | 60.39± 5.02 | 74.87± 4.35 | 81.5 ± 3.55 | 86.12± 3.04 |

Table 5: Average and standard deviation evaluation accuracy for main experiment (**A**).

| Optimizer | 10-Step | 25-Step | 50-Step | 100-Step | 250-Step | 500-Step | 1000-Step | 2500-Step |
|---|---|---|---|---|---|---|---|---|
| L3RS | 1.17 ± 0.13 | 0.82 ± 0.11 | 0.67 ± 0.10 | 0.56 ± 0.10 | 0.46 ± 0.09 | 0.39 ± 0.08 | 0.38 ± 0.09 | 0.43 ± 0.11 |
| VeLO (ft) | 1.35 ± 0.17 | 0.90 ± 0.13 | 0.75 ± 0.11 | 0.62 ± 0.11 | 0.50 ± 0.09 | 0.45 ± 0.10 | 0.92 ± 0.96 | 3.23 ± 0.12 |
| VeLO (og) | 1.75 ± 0.18 | 1.81 ± 0.16 | 1.57 ± 0.14 | 1.33 ± 0.14 | 0.98 ± 0.13 | 0.74 ± 0.12 | 0.61 ± 0.11 | 0.67 ± 0.14 |
| Adam (cosine), 1e-3 | 1.67 ± 0.15 | 1.20 ± 0.14 | 1.01 ± 0.13 | 0.82 ± 0.13 | 0.61 ± 0.11 | 0.50 ± 0.10 | 0.45 ± 0.09 | 0.49 ± 0.13 |
| Adam (cosine), 1e-4 | 3.18 ± 0.06 | 2.63 ± 0.09 | 1.94 ± 0.13 | 1.23 ± 0.14 | 0.69 ± 0.11 | 0.51 ± 0.10 | 0.42 ± 0.09 | 0.40 ± 0.10 |
| Adam (cosine), 1e-5 | 3.56 ± 0.05 | 3.49 ± 0.05 | 3.38 ± 0.05 | 3.17 ± 0.06 | 2.60 ± 0.09 | 1.88 ± 0.13 | 1.17 ± 0.14 | 0.66 ± 0.11 |
| Adam (const), 1e-3 | 1.44 ± 0.15 | 1.34 ± 0.14 | 1.20 ± 0.15 | 1.00 ± 0.13 | 0.83 ± 0.12 | 0.72 ± 0.12 | 0.66 ± 0.11 | 0.70 ± 0.15 |
| Adam (const), 1e-4 | 2.80 ± 0.08 | 1.95 ± 0.13 | 1.24 ± 0.14 | 0.80 ± 0.12 | 0.56 ± 0.10 | 0.46 ± 0.09 | 0.43 ± 0.09 | 0.46 ± 0.12 |
| Adam (const), 1e-5 | 3.51 ± 0.05 | 3.38 ± 0.05 | 3.17 ± 0.06 | 2.78 ± 0.08 | 1.89 ± 0.13 | 1.17 ± 0.14 | 0.74 ± 0.11 | 0.50 ± 0.10 |
| Adabelief, 1e-3 | 1.53 ± 0.15 | 1.45 ± 0.15 | 1.25 ± 0.15 | 1.04 ± 0.13 | 0.85 ± 0.13 | 0.74 ± 0.12 | 0.67 ± 0.11 | 0.7 ± 0.15 |
| Adabelief, 1e-4 | 2.68 ± 0.09 | 1.76 ± 0.13 | 1.11 ± 0.14 | 0.76 ± 0.12 | 0.55 ± 0.1 | 0.45 ± 0.09 | 0.43 ± 0.1 | 0.48 ± 0.12 |
| Adabelief, 1e-5 | 3.5 ± 0.05 | 3.33 ± 0.05 | 3.07 ± 0.07 | 2.6 ± 0.09 | 1.66 ± 0.14 | 1.03 ± 0.13 | 0.68 ± 0.11 | 0.48 ± 0.1 |

Table 6: Average and standard deviation evaluation loss for main experiment (**A**).

| Optimizer | 10-Step | 25-Step | 50-Step | 100-Step | 250-Step | 500-Step | 1000-Step | 2500-Step |
|---|---|---|---|---|---|---|---|---|
| VeLO (og) | 42.53± 4.73 | 67.93± 4.4 | 73.68± 3.65 | 76.35± 3.58 | 79.12± 3.45 | 80.71± 3.2 | 81.36± 3.11 | 81.6 ± 3.23 |
| Adam (cosine), 1e-3 | 7.98 ± 2.16 | 16.93± 3.14 | 34.58± 4.39 | 54.65± 5.05 | 69.84± 4.55 | 74.95± 4.0 | 78.0 ± 3.68 | 80.52± 3.41 |
| Adam (cosine), 1e-4 | 4.18 ± 1.47 | 4.53 ± 1.59 | 5.32 ± 1.7 | 7.52 ± 2.06 | 16.86± 3.24 | 34.34± 4.53 | 54.8 ± 4.99 | 70.14± 4.72 |
| Adam (cosine), 1e-5 | 3.95 ± 1.38 | 3.98 ± 1.38 | 4.05 ± 1.39 | 4.21 ± 1.45 | 4.55 ± 1.54 | 5.24 ± 1.55 | 7.32 ± 1.96 | 16.74± 3.15 |
| Adam (const), 1e-3 | 13.31± 2.71 | 33.5 ± 4.31 | 53.59± 4.81 | 65.91± 4.67 | 74.17± 4.07 | 77.34± 3.67 | 79.53± 3.51 | 81.17± 3.45 |
| Adam (const), 1e-4 | 4.44 ± 1.51 | 5.22 ± 1.59 | 7.4 ± 1.97 | 13.27± 3.03 | 34.17± 4.59 | 54.5 ± 4.94 | 67.28± 4.72 | 75.17± 3.93 |
| Adam (const), 1e-5 | 3.98 ± 1.38 | 4.01 ± 1.41 | 4.21 ± 1.43 | 4.43 ± 1.52 | 5.22 ± 1.57 | 7.32 ± 2.0 | 13.13± 3.05 | 34.35± 4.57 |

Table 7: Average and standard deviation evaluation accuracy for all Head-only fine-tuning from main experiment (**A**).

| Optimizer | 10-Step | 25-Step | 50-Step | 100-Step | 250-Step | 500-Step | 1000-Step | 2500-Step |
|---|---|---|---|---|---|---|---|---|
| VeLO (og) | 2.21 ± 0.11 | 1.11 ± 0.14 | 0.92 ± 0.14 | 0.85 ± 0.13 | 0.76 ± 0.13 | 0.68 ± 0.12 | 0.65 ± 0.12 | 0.66 ± 0.12 |
| Adam (cosine), 1e-3 | 3.20 ± 0.05 | 2.88 ± 0.07 | 2.45 ± 0.09 | 1.86 ± 0.11 | 1.21 ± 0.12 | 0.93 ± 0.12 | 0.77 ± 0.11 | 0.66 ± 0.11 |
| Adam (cosine), 1e-4 | 3.55 ± 0.05 | 3.48 ± 0.05 | 3.38 ± 0.05 | 3.23 ± 0.05 | 2.89 ± 0.06 | 2.46 ± 0.08 | 1.88 ± 0.11 | 1.21 ± 0.12 |
| Adam (cosine), 1e-5 | 3.60 ± 0.05 | 3.59 ± 0.05 | 3.58 ± 0.05 | 3.55 ± 0.05 | 3.48 ± 0.05 | 3.39 ± 0.05 | 3.23 ± 0.05 | 2.90 ± 0.06 |
| Adam (const), 1e-3 | 3.00 ± 0.06 | 2.47 ± 0.09 | 1.87 ± 0.11 | 1.35 ± 0.12 | 0.95 ± 0.12 | 0.79 ± 0.11 | 0.70 ± 0.11 | 0.64 ± 0.11 |
| Adam (const), 1e-4 | 3.50 ± 0.05 | 3.38 ± 0.05 | 3.23 ± 0.05 | 3.00 ± 0.06 | 2.47 ± 0.08 | 1.89 ± 0.11 | 1.35 ± 0.12 | 0.93 ± 0.12 |
| Adam (const), 1e-5 | 3.59 ± 0.05 | 3.58 ± 0.05 | 3.55 ± 0.05 | 3.51 ± 0.05 | 3.39 ± 0.05 | 3.24 ± 0.05 | 3.00 ± 0.06 | 2.46 ± 0.08 |

Table 8: Average and standard deviation evaluation loss for all Head-only fine-tuning from main experiment (**A**).

| Optimizer | 10-Step | 25-Step | 50-Step | 100-Step | 250-Step | 500-Step | 1000-Step | 2500-Step |
|---|---|---|---|---|---|---|---|---|
| L3RS | 59.2 ± 3.87 | 69.16± 3.32 | 73.64± 3.23 | 76.75± 3.32 | 80.67± 3.07 | 82.74± 2.92 | 84.32± 2.82 | 85.51± 2.81 |
| VeLO (ft) | 50.13± 7.32 | 64.78± 3.46 | 70.46± 3.29 | 74.1 ± 7.83 | 79.12± 3.21 | 76.54± 12.9 | 56.13± 30.47 | 4.02 ± 0.36 |
| VeLO (og) | 43.13± 3.98 | 44.58± 4.18 | 51.26± 4.18 | 59.02± 4.0 | 68.38± 3.83 | 75.14± 3.47 | 79.97± 3.49 | 83.07± 3.13 |
| Adam (cosine), 1e-3 | 48.85± 3.79 | 60.39± 3.34 | 66.11± 3.59 | 71.81± 3.64 | 77.71± 3.44 | 80.73± 3.17 | 82.99± 2.96 | 84.97± 2.76 |
| Adam (cosine), 1e-4 | 9.87 ± 2.24 | 23.3 ± 2.97 | 44.16± 3.99 | 63.11± 3.72 | 74.9 ± 3.65 | 79.71± 3.0 | 82.68± 3.04 | 85.17± 2.98 |
| Adam (cosine), 1e-5 | 4.38 ± 1.53 | 4.98 ± 1.55 | 6.07 ± 1.68 | 9.06 ± 2.06 | 22.83± 3.06 | 45.84± 3.56 | 65.3 ± 3.72 | 76.39± 3.54 |
| Adam (const), 1e-3 | 52.54± 4.3 | 56.19± 3.93 | 61.15± 3.92 | 65.87± 3.94 | 71.71± 3.52 | 74.83± 3.42 | 77.42± 3.57 | 80.51± 3.28 |
| Adam (const), 1e-4 | 17.32± 2.81 | 43.3 ± 3.74 | 62.03± 3.48 | 71.65± 3.48 | 78.01± 3.28 | 80.57± 2.85 | 82.72± 3.01 | 84.66± 2.97 |
| Adam (const), 1e-5 | 4.73 ± 1.54 | 6.08 ± 1.66 | 9.05 ± 2.15 | 17.72± 2.64 | 45.84± 3.79 | 64.93± 3.85 | 74.3 ± 3.65 | 80.72± 3.3 |
| Adabelief, 1e-3 | 50.39± 4.24 | 54.3 ± 3.83 | 59.78± 3.84 | 65.5 ± 3.8 | 71.14± 3.73 | 74.35± 3.47 | 77.27± 3.43 | 80.31± 3.45 |
| Adabelief, 1e-4 | 20.61± 3.14 | 48.45± 3.95 | 64.25± 3.46 | 72.61± 3.59 | 78.22± 3.28 | 80.68± 2.81 | 82.59± 3.01 | 84.45± 2.83 |
| Adabelief, 1e-5 | 4.87 ± 1.54 | 6.61 ± 1.72 | 10.96± 2.26 | 22.72± 3.09 | 52.43± 3.87 | 67.95± 3.63 | 75.61± 3.73 | 81.28± 3.23 |

Table 9: Average and standard deviation evaluation accuracy for main experiment (**B**).

| Optimizer | 10-Step | 25-Step | 50-Step | 100-Step | 250-Step | 500-Step | 1000-Step | 2500-Step |
|---|---|---|---|---|---|---|---|---|
| L3RS | 1.44 ± 0.10 | 1.02 ± 0.09 | 0.85 ± 0.09 | 0.73 ± 0.10 | 0.61 ± 0.09 | 0.55 ± 0.08 | 0.49 ± 0.08 | 0.45 ± 0.08 |
| VeLO (ft) | 1.73 ± 0.25 | 1.13 ± 0.10 | 0.95 ± 0.09 | 0.82 ± 0.26 | 0.66 ± 0.09 | 0.78 ± 0.51 | 3.65 ± 15.11 | 1209.28± 11410.45 |
| VeLO (og) | 1.88 ± 0.12 | 1.80 ± 0.12 | 1.57 ± 0.12 | 1.34 ± 0.10 | 1.01 ± 0.11 | 0.79 ± 0.10 | 0.64 ± 0.09 | 0.53 ± 0.08 |
| Adam (cosine), 1e-3 | 1.82 ± 0.10 | 1.33 ± 0.09 | 1.11 ± 0.10 | 0.91 ± 0.10 | 0.71 ± 0.09 | 0.61 ± 0.08 | 0.54 ± 0.08 | 0.47 ± 0.08 |
| Adam (cosine), 1e-4 | 3.19 ± 0.06 | 2.75 ± 0.06 | 2.16 ± 0.08 | 1.42 ± 0.09 | 0.85 ± 0.09 | 0.66 ± 0.09 | 0.55 ± 0.08 | 0.46 ± 0.08 |
| Adam (cosine), 1e-5 | 3.52 ± 0.06 | 3.46 ± 0.06 | 3.37 ± 0.06 | 3.21 ± 0.06 | 2.75 ± 0.06 | 2.10 ± 0.09 | 1.35 ± 0.10 | 0.80 ± 0.09 |
| Adam (const), 1e-3 | 1.61 ± 0.12 | 1.45 ± 0.11 | 1.26 ± 0.10 | 1.09 ± 0.11 | 0.90 ± 0.10 | 0.80 ± 0.10 | 0.71 ± 0.10 | 0.61 ± 0.09 |
| Adam (const), 1e-4 | 2.91 ± 0.06 | 2.18 ± 0.08 | 1.46 ± 0.09 | 0.98 ± 0.10 | 0.72 ± 0.09 | 0.61 ± 0.08 | 0.55 ± 0.08 | 0.48 ± 0.08 |
| Adam (const), 1e-5 | 3.48 ± 0.06 | 3.38 ± 0.06 | 3.21 ± 0.06 | 2.90 ± 0.06 | 2.10 ± 0.09 | 1.36 ± 0.10 | 0.90 ± 0.09 | 0.63 ± 0.09 |
| Adabelief, 1e-3 | 1.68 ± 0.13 | 1.54 ± 0.11 | 1.31 ± 0.1 | 1.11 ± 0.11 | 0.93 ± 0.11 | 0.81 ± 0.1 | 0.72 ± 0.09 | 0.62 ± 0.09 |
| Adabelief, 1e-4 | 2.82 ± 0.06 | 2.01 ± 0.09 | 1.32 ± 0.09 | 0.93 ± 0.1 | 0.7 ± 0.09 | 0.61 ± 0.08 | 0.55 ± 0.08 | 0.48 ± 0.08 |
| Adabelief, 1e-5 | 3.47 ± 0.06 | 3.34 ± 0.06 | 3.13 ± 0.06 | 2.76 ± 0.07 | 1.88 ± 0.09 | 1.21 ± 0.1 | 0.84 ± 0.09 | 0.61 ± 0.09 |

Table 10: Average and standard deviation evaluation loss for main experiment (**B**).

| Optimizer | 10-Step | 25-Step | 50-Step | 100-Step | 250-Step | 500-Step | 1000-Step | 2500-Step |
|---|---|---|---|---|---|---|---|---|
| L3RS | 65.65± 4.61 | 73.13± 3.99 | 77.0 ± 4.02 | 80.33± 3.59 | 83.79± 3.06 | 85.93± 2.97 | 86.91± 2.78 | 87.04± 2.68 |
| VeLO (ft) | 57.85± 7.39 | 69.6 ± 4.0 | 73.51± 4.09 | 77.96± 3.8 | 82.18± 3.22 | 84.46± 2.95 | 82.67± 11.37 | 4.04 ± 0.27 |
| VeLO (og) | 52.35± 4.67 | 49.56± 4.78 | 54.52± 4.72 | 61.44± 4.58 | 71.37± 4.1 | 77.87± 3.67 | 81.7 ± 3.15 | 82.82± 3.08 |
| Adam (cosine), 1e-3 | 55.36± 4.8 | 62.93± 4.44 | 67.45± 4.35 | 72.54± 4.18 | 79.14± 3.41 | 82.61± 3.09 | 84.43± 3.19 | 85.36± 3.17 |
| Adam (cosine), 1e-4 | 13.27± 2.7 | 33.25± 4.39 | 54.11± 4.84 | 68.21± 4.46 | 78.83± 3.61 | 83.49± 3.16 | 85.96± 2.98 | 87.31± 2.66 |
| Adam (cosine), 1e-5 | 4.37 ± 1.52 | 5.26 ± 1.72 | 7.14 ± 2.02 | 12.29± 2.7 | 33.43± 4.16 | 56.37± 4.73 | 70.46± 4.56 | 80.28± 3.51 |
| Adam (const), 1e-3 | 56.2 ± 4.9 | 58.38± 4.31 | 61.7 ± 4.4 | 66.41± 4.45 | 72.24± 4.05 | 75.95± 3.68 | 78.45± 3.38 | 80.04± 3.6 |
| Adam (const), 1e-4 | 25.4 ± 3.79 | 53.72± 4.75 | 67.04± 4.56 | 75.64± 4.01 | 81.56± 3.42 | 84.25± 3.09 | 85.96± 2.8 | 85.92± 2.75 |
| Adam (const), 1e-5 | 4.87 ± 1.58 | 6.97 ± 2.01 | 12.38± 2.59 | 26.29± 4.03 | 56.09± 4.69 | 70.48± 4.56 | 78.41± 3.75 | 83.96± 3.11 |
| Adabelief, 1e-3 | 54.03± 4.92 | 55.73± 4.49 | 59.86± 4.62 | 65.2 ± 4.01 | 71.07± 4.25 | 75.6 ± 3.85 | 78.37± 3.3 | 79.71± 3.63 |
| Adabelief, 1e-4 | 30.27± 4.26 | 57.77± 4.66 | 68.92± 4.48 | 76.41± 3.89 | 81.81± 3.46 | 84.25± 3.18 | 85.87± 2.75 | 85.6 ± 3.08 |
| Adabelief, 1e-5 | 5.13 ± 1.62 | 7.96 ± 2.15 | 15.48± 3.09 | 33.73± 4.38 | 61.44± 4.93 | 72.96± 4.24 | 79.71± 3.66 | 84.35± 3.06 |

Table 11: Average and standard deviation evaluation accuracy for main experiment (**C**).

| Optimizer | 10-Step | 25-Step | 50-Step | 100-Step | 250-Step | 500-Step | 1000-Step | 2500-Step |
|---|---|---|---|---|---|---|---|---|
| L3RS | 1.34 ± 0.13 | 0.95 ± 0.12 | 0.78 ± 0.11 | 0.66 ± 0.11 | 0.54 ± 0.10 | 0.47 ± 0.09 | 0.44 ± 0.09 | 0.50 ± 0.11 |
| VeLO (ft) | 1.64 ± 0.27 | 1.02 ± 0.13 | 0.88 ± 0.13 | 0.73 ± 0.12 | 0.59 ± 0.10 | 0.52 ± 0.09 | 0.58 ± 0.40 | 15.24± 39.84 |
| VeLO (og) | 1.66 ± 0.16 | 1.73 ± 0.15 | 1.55 ± 0.15 | 1.31 ± 0.14 | 0.97 ± 0.14 | 0.75 ± 0.12 | 0.62 ± 0.11 | 0.71 ± 0.15 |
| Adam (cosine), 1e-3 | 1.75 ± 0.13 | 1.33 ± 0.15 | 1.14 ± 0.13 | 0.95 ± 0.13 | 0.72 ± 0.11 | 0.60 ± 0.10 | 0.52 ± 0.11 | 0.55 ± 0.13 |
| Adam (cosine), 1e-4 | 3.05 ± 0.05 | 2.63 ± 0.07 | 2.05 ± 0.10 | 1.36 ± 0.13 | 0.79 ± 0.11 | 0.59 ± 0.10 | 0.48 ± 0.09 | 0.46 ± 0.10 |
| Adam (cosine), 1e-5 | 3.35 ± 0.05 | 3.30 ± 0.05 | 3.22 ± 0.05 | 3.07 ± 0.05 | 2.63 ± 0.07 | 1.98 ± 0.11 | 1.28 ± 0.12 | 0.74 ± 0.11 |
| Adam (const), 1e-3 | 1.60 ± 0.16 | 1.48 ± 0.14 | 1.33 ± 0.15 | 1.15 ± 0.13 | 0.94 ± 0.13 | 0.81 ± 0.12 | 0.74 ± 0.12 | 0.76 ± 0.16 |
| Adam (const), 1e-4 | 2.79 ± 0.06 | 2.08 ± 0.10 | 1.39 ± 0.12 | 0.93 ± 0.12 | 0.65 ± 0.10 | 0.54 ± 0.09 | 0.49 ± 0.09 | 0.55 ± 0.12 |
| Adam (const), 1e-5 | 3.32 ± 0.05 | 3.22 ± 0.05 | 3.07 ± 0.05 | 2.77 ± 0.06 | 1.99 ± 0.11 | 1.28 ± 0.12 | 0.83 ± 0.11 | 0.56 ± 0.10 |
| Adabelief, 1e-3 | 1.67 ± 0.16 | 1.57 ± 0.15 | 1.38 ± 0.15 | 1.19 ± 0.13 | 0.97 ± 0.13 | 0.83 ± 0.12 | 0.75 ± 0.13 | 0.78 ± 0.16 |
| Adabelief, 1e-4 | 2.7 ± 0.07 | 1.9 ± 0.11 | 1.25 ± 0.13 | 0.87 ± 0.12 | 0.64 ± 0.1 | 0.54 ± 0.09 | 0.49 ± 0.09 | 0.56 ± 0.13 |
| Adabelief, 1e-5 | 3.31 ± 0.05 | 3.19 ± 0.04 | 2.99 ± 0.05 | 2.62 ± 0.07 | 1.75 ± 0.12 | 1.13 ± 0.12 | 0.76 ± 0.11 | 0.54 ± 0.09 |

Table 12: Average and standard deviation evaluation loss for main experiment (**C**).

| Optimizer | 10-Step | 25-Step | 50-Step | 100-Step | 250-Step | 500-Step | 1000-Step | 2500-Step |
|---|---|---|---|---|---|---|---|---|
| L3RS | 72.15± 3.71 | 77.51± 3.11 | 80.22± 3.35 | 81.76± 3.27 | 83.89± 3.0 | 84.96± 2.81 | 85.95± 2.83 | 87.06± 2.79 |
| VeLO (ft) | 66.93± 7.24 | 75.68± 3.35 | 78.26± 3.38 | 80.88± 2.98 | 83.33± 3.06 | 80.12± 12.15 | 59.12± 30.85 | 4.05 ± 0.37 |
| VeLO (og) | 58.38± 4.31 | 54.52± 3.97 | 58.73± 4.2 | 64.48± 3.83 | 73.04± 3.49 | 78.45± 3.33 | 81.84± 3.41 | 84.44± 2.41 |
| Adam (cosine), 1e-3 | 60.65± 4.08 | 67.66± 3.13 | 70.77± 3.74 | 74.73± 3.69 | 79.2 ± 3.24 | 81.41± 3.25 | 83.47± 3.13 | 85.58± 3.78 |
| Adam (cosine), 1e-4 | 15.17± 2.9 | 38.45± 4.37 | 61.08± 3.91 | 74.22± 3.33 | 80.98± 3.45 | 83.73± 3.03 | 85.26± 3.0 | 86.8 ± 2.98 |
| Adam (cosine), 1e-5 | 4.86 ± 1.84 | 5.88 ± 2.0 | 7.95 ± 2.25 | 13.78± 3.02 | 38.75± 4.17 | 63.86± 3.64 | 76.06± 3.29 | 82.1 ± 2.81 |
| Adam (const), 1e-3 | 60.29± 4.24 | 60.9 ± 3.63 | 63.8 ± 3.57 | 68.15± 3.88 | 72.46± 3.92 | 75.25± 3.52 | 77.68± 3.61 | 80.7 ± 3.14 |
| Adam (const), 1e-4 | 29.37± 4.04 | 60.46± 3.96 | 73.09± 3.48 | 78.62± 3.34 | 81.95± 3.29 | 83.27± 2.93 | 84.38± 2.99 | 85.86± 2.64 |
| Adam (const), 1e-5 | 5.43 ± 1.9 | 7.8 ± 2.26 | 13.8 ± 2.96 | 30.14± 4.13 | 63.53± 3.66 | 75.85± 3.25 | 80.89± 3.24 | 84.49± 3.66 |
| Adabelief, 1e-3 | 57.99± 4.46 | 58.62± 3.77 | 62.43± 3.71 | 67.29± 3.94 | 72.01± 3.72 | 74.49± 3.69 | 77.36± 3.42 | 80.34± 3.55 |
| Adabelief, 1e-4 | 35.2 ± 4.21 | 64.75± 3.66 | 74.73± 3.42 | 79.11± 3.33 | 82.03± 3.35 | 83.36± 2.84 | 84.28± 2.93 | 85.5 ± 2.89 |
| Adabelief, 1e-5 | 5.71 ± 1.94 | 9.06 ± 2.36 | 18.14± 3.37 | 40.06± 4.47 | 69.3 ± 3.58 | 77.89± 3.31 | 81.77± 3.35 | 84.79± 3.13 |

Table 13: Average and standard deviation evaluation accuracy for main experiment (**D**).

| Optimizer | 10-Step | 25-Step | 50-Step | 100-Step | 250-Step | 500-Step | 1000-Step | 2500-Step |
|---|---|---|---|---|---|---|---|---|
| L3RS | 1.07 ± 0.09 | 0.76 ± 0.08 | 0.65 ± 0.09 | 0.58 ± 0.09 | 0.51 ± 0.08 | 0.47 ± 0.08 | 0.43 ± 0.07 | 0.4 ± 0.08 |
| VeLO (ft) | 1.22 ± 0.25 | 0.77 ± 0.09 | 0.69 ± 0.09 | 0.6 ± 0.09 | 0.52 ± 0.08 | 1.29 ± 3.69 | 3.53 ± 12.36 | 2.5e4± 1.6e5 |
| VeLO (og) | 2.85 ± 0.09 | 2.72 ± 0.10 | 2.60 ± 0.11 | 2.41 ± 0.13 | 1.95 ± 0.15 | 1.57 ± 0.14 | 1.22 ± 0.14 | 0.95 ± 0.15 |
| Adam (cosine), 1e-3 | 2.98 ± 0.06 | 2.81 ± 0.08 | 2.67 ± 0.09 | 2.50 ± 0.11 | 2.13 ± 0.14 | 1.84 ± 0.14 | 1.52 ± 0.14 | 1.13 ± 0.13 |
| Adam (cosine), 1e-4 | 3.26 ± 0.03 | 3.18 ± 0.03 | 3.08 ± 0.04 | 2.93 ± 0.06 | 2.68 ± 0.08 | 2.42 ± 0.11 | 2.00 ± 0.13 | 1.49 ± 0.13 |
| Adam (cosine), 1e-5 | 3.31 ± 0.03 | 3.30 ± 0.03 | 3.29 ± 0.03 | 3.26 ± 0.03 | 3.18 ± 0.03 | 3.07 ± 0.04 | 2.91 ± 0.06 | 2.65 ± 0.09 |
| Adam (const), 1e-3 | 2.88 ± 0.07 | 2.73 ± 0.09 | 2.61 ± 0.10 | 2.44 ± 0.12 | 2.14 ± 0.14 | 1.86 ± 0.15 | 1.56 ± 0.14 | 1.20 ± 0.15 |
| Adam (const), 1e-4 | 3.21 ± 0.03 | 3.08 ± 0.04 | 2.93 ± 0.06 | 2.76 ± 0.08 | 2.46 ± 0.11 | 2.10 ± 0.12 | 1.76 ± 0.14 | 1.37 ± 0.14 |
| Adam (const), 1e-5 | 3.31 ± 0.03 | 3.29 ± 0.03 | 3.26 ± 0.03 | 3.21 ± 0.03 | 3.07 ± 0.04 | 2.91 ± 0.06 | 2.72 ± 0.08 | 2.36 ± 0.11 |
| Adabelief, 1e-3 | 1.45 ± 0.12 | 1.38 ± 0.11 | 1.22 ± 0.1 | 1.06 ± 0.11 | 0.9 ± 0.1 | 0.81 ± 0.1 | 0.71 ± 0.09 | 0.62 ± 0.09 |
| Adabelief, 1e-4 | 2.58 ± 0.07 | 1.67 ± 0.09 | 1.01 ± 0.09 | 0.74 ± 0.08 | 0.59 ± 0.08 | 0.53 ± 0.08 | 0.49 ± 0.08 | 0.45 ± 0.08 |
| Adabelief, 1e-5 | 3.29 ± 0.06 | 3.15 ± 0.06 | 2.91 ± 0.06 | 2.48 ± 0.07 | 1.49 ± 0.09 | 0.9 ± 0.08 | 0.64 ± 0.08 | 0.5 ± 0.08 |

Table 14: Average and standard deviation evaluation loss for main experiment (**D**).

| Optimizer | 10-Step | 25-Step | 50-Step | 100-Step | 250-Step | 500-Step | 1000-Step | 2500-Step |
|---|---|---|---|---|---|---|---|---|
| L3RS | 15.77± 2.99 | 21.11± 3.52 | 25.5 ± 3.88 | 30.46± 4.05 | 40.32± 4.33 | 47.69± 4.26 | 54.74± 4.09 | 60.9 ± 4.04 |
| VeLO (ft) | 16.18± 3.69 | 18.59± 3.27 | 18.55± 6.33 | 11.59± 8.62 | 5.49 ± 5.06 | 7.87 ± 7.24 | 4.38 ± 0.98 | 3.98 ± 0.27 |
| VeLO (og) | 17.69± 3.1 | 21.04± 3.66 | 24.83± 3.72 | 30.11± 4.28 | 42.68± 4.34 | 53.73± 3.87 | 64.06± 4.14 | 73.21± 3.92 |
| Adam (cosine), 1e-3 | 14.48± 2.49 | 19.26± 3.13 | 23.45± 3.71 | 28.14± 3.71 | 38.59± 4.42 | 46.64± 4.2 | 55.25± 4.19 | 66.46± 3.88 |
| Adam (cosine), 1e-4 | 5.02 ± 1.43 | 7.46 ± 1.92 | 12.07± 2.49 | 17.92± 3.09 | 24.83± 3.62 | 32.58± 4.26 | 43.25± 4.41 | 56.57± 3.98 |
| Adam (cosine), 1e-5 | 4.17 ± 1.18 | 4.29 ± 1.2 | 4.46 ± 1.25 | 4.97 ± 1.35 | 7.2 ± 1.96 | 12.18± 2.64 | 18.75± 3.21 | 26.31± 3.67 |
| Adam (const), 1e-3 | 17.19± 3.05 | 20.19± 3.34 | 23.95± 3.68 | 28.74± 4.2 | 37.17± 4.62 | 44.95± 4.21 | 53.75± 4.12 | 65.26± 3.64 |
| Adam (const), 1e-4 | 6.34 ± 1.72 | 12.18± 2.62 | 17.68± 2.95 | 22.36± 3.41 | 30.64± 4.12 | 40.09± 4.14 | 48.52± 4.22 | 59.35± 4.15 |
| Adam (const), 1e-5 | 4.24 ± 1.23 | 4.43 ± 1.24 | 4.95 ± 1.38 | 6.37 ± 1.64 | 12.21± 2.6 | 18.84± 3.19 | 24.37± 3.61 | 34.44± 4.33 |
| Adabelief, 1e-3 | 17.32± 3.14 | 20.28± 3.38 | 23.84± 3.85 | 28.66± 4.05 | 37.32± 4.57 | 44.78± 4.12 | 53.8 ± 3.87 | 65.44± 3.91 |
| Adabelief, 1e-4 | 7.19 ± 1.89 | 13.93± 2.94 | 19.1 ± 3.03 | 23.64± 3.63 | 32.51± 4.29 | 41.35± 4.23 | 49.64± 4.2 | 60.12± 4.18 |
| Adabelief, 1e-5 | 4.26 ± 1.22 | 4.58 ± 1.2 | 5.48 ± 1.53 | 7.8 ± 1.96 | 15.59± 3.15 | 21.24± 3.23 | 26.93± 3.89 | 38.28± 4.38 |

Table 15: Average and standard deviation evaluation accuracy for main experiment (**E**).

| Optimizer | 10-Step | 25-Step | 50-Step | 100-Step | 250-Step | 500-Step | 1000-Step | 2500-Step |
|---|---|---|---|---|---|---|---|---|
| L3RS | 1.07 ± 0.09 | 0.76 ± 0.08 | 0.65 ± 0.09 | 0.58 ± 0.09 | 0.51 ± 0.08 | 0.47 ± 0.08 | 0.43 ± 0.07 | 0.40 ± 0.08 |
| VeLO (ft) | 1.22 ± 0.25 | 0.77 ± 0.09 | 0.69 ± 0.09 | 0.60 ± 0.09 | 0.52 ± 0.08 | 1.29 ± 3.69 | 3.53 ± 12.36 | 2.5e4± 1.6e5 |
| VeLO (og) | 1.38 ± 0.12 | 1.49 ± 0.12 | 1.33 ± 0.11 | 1.14 ± 0.11 | 0.85 ± 0.10 | 0.69 ± 0.09 | 0.57 ± 0.09 | 0.49 ± 0.08 |
| Adam (cosine), 1e-3 | 1.52 ± 0.10 | 1.11 ± 0.08 | 0.97 ± 0.10 | 0.82 ± 0.10 | 0.67 ± 0.09 | 0.58 ± 0.09 | 0.52 ± 0.08 | 0.46 ± 0.08 |
| Adam (cosine), 1e-4 | 2.98 ± 0.06 | 2.50 ± 0.07 | 1.85 ± 0.08 | 1.12 ± 0.09 | 0.67 ± 0.08 | 0.54 ± 0.08 | 0.47 ± 0.08 | 0.41 ± 0.07 |
| Adam (cosine), 1e-5 | 3.34 ± 0.06 | 3.28 ± 0.06 | 3.19 ± 0.06 | 3.01 ± 0.06 | 2.50 ± 0.07 | 1.78 ± 0.08 | 1.05 ± 0.08 | 0.63 ± 0.08 |
| Adam (const), 1e-3 | 1.38 ± 0.12 | 1.30 ± 0.11 | 1.17 ± 0.10 | 1.03 ± 0.11 | 0.88 ± 0.11 | 0.79 ± 0.09 | 0.70 ± 0.09 | 0.61 ± 0.10 |
| Adam (const), 1e-4 | 2.69 ± 0.07 | 1.89 ± 0.08 | 1.15 ± 0.09 | 0.77 ± 0.08 | 0.60 ± 0.08 | 0.53 ± 0.08 | 0.49 ± 0.08 | 0.45 ± 0.08 |
| Adam (const), 1e-5 | 3.30 ± 0.06 | 3.19 ± 0.06 | 3.01 ± 0.06 | 2.67 ± 0.07 | 1.78 ± 0.08 | 1.05 ± 0.08 | 0.69 ± 0.08 | 0.51 ± 0.08 |
| Adabelief, 1e-3 | 2.87 ± 0.08 | 2.73 ± 0.1 | 2.61 ± 0.1 | 2.44 ± 0.12 | 2.14 ± 0.14 | 1.86 ± 0.14 | 1.56 ± 0.14 | 1.19 ± 0.14 |
| Adabelief, 1e-4 | 3.19 ± 0.03 | 3.03 ± 0.04 | 2.88 ± 0.07 | 2.7 ± 0.09 | 2.38 ± 0.12 | 2.04 ± 0.13 | 1.72 ± 0.14 | 1.35 ± 0.14 |
| Adabelief, 1e-5 | 3.3 ± 0.03 | 3.28 ± 0.03 | 3.24 ± 0.03 | 3.17 ± 0.03 | 3.0 ± 0.05 | 2.83 ± 0.07 | 2.63 ± 0.09 | 2.2 ± 0.13 |

Table 16: Average and standard deviation evaluation loss for main experiment (**E**).

| Optimizer | 10-Step | 25-Step | 50-Step | 100-Step | 250-Step | 500-Step | 1000-Step | 2500-Step |
|---|---|---|---|---|---|---|---|---|
| L3RS | 65.81± 6.6 | 73.91± 5.47 | 77.54± 5.33 | 81.52± 3.87 | 84.53± 3.5 | 85.9 ± 3.14 | 87.58± 3.48 | 88.63± 3.06 |
| VeLO (ft) | 53.4 ± 5.55 | 63.32± 5.77 | 72.23± 4.67 | 78.44± 4.34 | 82.34± 3.37 | 84.61± 4.01 | 12.07± 23.98 | 3.98 ± 0.16 |
| VeLO (og) | 48.87± 4.9 | 48.52± 5.32 | 54.84± 5.56 | 61.05± 3.49 | 70.55± 4.55 | 77.5 ± 4.21 | 81.76± 4.3 | 83.71± 3.34 |
| Adam (cosine), 1e-3 | 55.62± 6.39 | 66.98± 6.22 | 72.54± 6.11 | 77.03± 4.55 | 82.11± 4.25 | 84.69± 4.09 | 86.64± 3.89 | 87.38± 3.37 |
| Adam (cosine), 1e-4 | 9.41 ± 2.06 | 22.07± 3.55 | 43.71± 6.21 | 64.69± 7.0 | 77.84± 5.46 | 82.19± 4.23 | 85.43± 3.76 | 87.46± 4.42 |
| Adam (cosine), 1e-5 | 4.3 ± 1.02 | 4.57 ± 1.22 | 5.66 ± 1.46 | 8.52 ± 2.04 | 21.33± 4.22 | 44.02± 6.52 | 64.77± 6.31 | 78.01± 5.37 |
| Adam (const), 1e-3 | 61.6 ± 5.85 | 66.72± 6.29 | 70.0 ± 4.76 | 71.56± 4.38 | 76.76± 3.78 | 81.41± 5.21 | 80.51± 3.76 | 82.5 ± 3.47 |
| Adam (const), 1e-4 | 17.03± 3.23 | 42.73± 6.26 | 62.81± 6.45 | 74.53± 5.75 | 81.59± 4.19 | 84.26± 4.01 | 86.64± 4.42 | 87.34± 3.32 |
| Adam (const), 1e-5 | 4.45 ± 1.12 | 5.66 ± 1.14 | 8.4 ± 1.97 | 16.84± 3.11 | 43.16± 6.55 | 64.48± 6.51 | 75.47± 5.58 | 82.7 ± 3.98 |

Table 17: Average and standard deviation evaluation accuracy for ResNet-18 experiment.

| Optimizer | 10-Step | 25-Step | 50-Step | 100-Step | 250-Step | 500-Step | 1000-Step | 2500-Step |
|---|---|---|---|---|---|---|---|---|
| L3RS | 1.24 ± 0.19 | 0.88 ± 0.18 | 0.74 ± 0.15 | 0.63 ± 0.13 | 0.52 ± 0.14 | 0.46 ± 0.13 | 0.43 ± 0.13 | 0.44 ± 0.14 |
| VeLO (ft) | 1.73 ± 0.17 | 1.21 ± 0.19 | 0.93 ± 0.16 | 0.7 ± 0.14 | 0.56 ± 0.12 | 0.52 ± 0.15 | 4.7e4± 1.1e5 | 9.3e5± 9.0e5 |
| VeLO (og) | 1.74 ± 0.15 | 1.79 ± 0.16 | 1.58 ± 0.17 | 1.35 ± 0.13 | 1.0 ± 0.16 | 0.75 ± 0.17 | 0.64 ± 0.16 | 0.7 ± 0.19 |
| Adam (cosine), 1e-3 | 1.7 ± 0.17 | 1.15 ± 0.2 | 0.93 ± 0.17 | 0.78 ± 0.14 | 0.59 ± 0.12 | 0.52 ± 0.13 | 0.47 ± 0.12 | 0.5 ± 0.15 |
| Adam (cosine), 1e-4 | 3.24 ± 0.04 | 2.78 ± 0.08 | 2.18 ± 0.13 | 1.49 ± 0.17 | 0.85 ± 0.15 | 0.62 ± 0.13 | 0.5 ± 0.11 | 0.43 ± 0.13 |
| Adam (cosine), 1e-5 | 3.57 ± 0.04 | 3.52 ± 0.04 | 3.43 ± 0.04 | 3.26 ± 0.03 | 2.8 ± 0.07 | 2.17 ± 0.12 | 1.46 ± 0.16 | 0.83 ± 0.15 |
| Adam (const), 1e-3 | 1.39 ± 0.18 | 1.17 ± 0.2 | 1.05 ± 0.17 | 0.98 ± 0.16 | 0.8 ± 0.13 | 0.69 ± 0.17 | 0.65 ± 0.14 | 0.71 ± 0.18 |
| Adam (const), 1e-4 | 2.96 ± 0.06 | 2.2 ± 0.13 | 1.5 ± 0.16 | 0.98 ± 0.17 | 0.65 ± 0.13 | 0.53 ± 0.13 | 0.46 ± 0.12 | 0.47 ± 0.15 |
| Adam (const), 1e-5 | 3.54 ± 0.04 | 3.43 ± 0.03 | 3.26 ± 0.04 | 2.95 ± 0.06 | 2.18 ± 0.12 | 1.46 ± 0.16 | 0.94 ± 0.16 | 0.61 ± 0.13 |

Table 18: Average and standard deviation evaluation loss for ResNet-18 experiment.

| Optimizer | 10-Step | 25-Step | 50-Step | 100-Step | 250-Step | 500-Step | 1000-Step | 2500-Step |
|---|---|---|---|---|---|---|---|---|
| L3RS | 60.62± 4.47 | 68.44± 3.71 | 72.54± 2.98 | 76.29± 2.81 | 80.0 ± 3.32 | 81.13± 3.02 | 82.34± 2.6 | 82.62± 2.86 |
| VeLO (ft) | 55.08± 3.43 | 66.33± 3.0 | 71.45± 3.47 | 74.61± 2.27 | 78.91± 3.61 | 79.26± 3.16 | 72.97± 9.93 | 5.27 ± 3.76 |
| VeLO (og) | 20.51± 5.35 | 30.55± 4.82 | 32.89± 4.34 | 39.53± 6.3 | 52.46± 4.55 | 63.79± 4.07 | 69.88± 3.88 | 71.25± 2.59 |
| Adam (cosine), 1e-3 | 28.05± 9.33 | 43.83± 5.17 | 49.96± 8.14 | 60.35± 4.44 | 69.69± 4.09 | 72.73± 3.13 | 74.18± 3.23 | 72.66± 3.92 |
| Adam (cosine), 1e-4 | 26.52± 2.95 | 49.34± 4.55 | 62.03± 3.61 | 70.2 ± 3.92 | 76.2 ± 3.01 | 79.45± 3.12 | 82.5 ± 2.14 | 82.62± 2.67 |
| Adam (cosine), 1e-5 | 5.27 ± 1.33 | 8.01 ± 1.25 | 12.85± 1.87 | 24.65± 3.27 | 52.93± 3.3 | 66.33± 3.83 | 73.01± 3.38 | 77.5 ± 3.43 |
| Adam (const), 1e-3 | 28.05± 10.93 | 39.79± 5.09 | 44.96± 6.1 | 49.77± 5.24 | 53.91± 5.5 | 58.67± 4.5 | 60.7 ± 4.23 | 63.52± 4.15 |
| Adam (const), 1e-4 | 40.82± 2.69 | 58.75± 5.46 | 66.72± 3.71 | 70.98± 3.45 | 75.7 ± 3.4 | 77.7 ± 2.49 | 79.06± 2.66 | 78.79± 2.36 |
| Adam (const), 1e-5 | 6.48 ± 1.69 | 12.77± 2.42 | 24.84± 2.6 | 45.74± 3.66 | 65.73± 3.77 | 72.62± 3.25 | 75.98± 3.43 | 79.14± 2.72 |

Table 19: Average and standard deviation evaluation accuracy for ViT experiment.

| Optimizer | 10-Step | 25-Step | 50-Step | 100-Step | 250-Step | 500-Step | 1000-Step | 2500-Step |
|---|---|---|---|---|---|---|---|---|
| L3RS | 1.36 ± 0.11 | 1.06 ± 0.1 | 0.9 ± 0.1 | 0.8 ± 0.1 | 0.67 ± 0.11 | 0.61 ± 0.1 | 0.64 ± 0.12 | 0.9 ± 0.19 |
| VeLO (ft) | 1.52 ± 0.12 | 1.13 ± 0.11 | 0.96 ± 0.1 | 0.83 ± 0.1 | 0.69 ± 0.11 | 0.68 ± 0.11 | 0.96 ± 0.35 | 85.23± 86.78 |
| VeLO (og) | 2.76 ± 0.13 | 2.42 ± 0.15 | 2.27 ± 0.17 | 2.0 ± 0.19 | 1.59 ± 0.14 | 1.23 ± 0.12 | 1.05 ± 0.15 | 1.17 ± 0.12 |
| Adam (cosine), 1e-3 | 2.63 ± 0.42 | 1.91 ± 0.19 | 1.67 ± 0.33 | 1.32 ± 0.15 | 1.02 ± 0.13 | 0.9 ± 0.11 | 0.88 ± 0.13 | 1.1 ± 0.18 |
| Adam (cosine), 1e-4 | 2.68 ± 0.08 | 1.84 ± 0.12 | 1.3 ± 0.13 | 1.02 ± 0.11 | 0.79 ± 0.1 | 0.68 ± 0.1 | 0.61 ± 0.09 | 0.68 ± 0.12 |
| Adam (cosine), 1e-5 | 3.59 ± 0.06 | 3.41 ± 0.06 | 3.13 ± 0.05 | 2.65 ± 0.07 | 1.73 ± 0.09 | 1.22 ± 0.1 | 0.94 ± 0.09 | 0.76 ± 0.1 |
| Adam (const), 1e-3 | 2.54 ± 0.4 | 2.06 ± 0.16 | 1.85 ± 0.19 | 1.65 ± 0.17 | 1.53 ± 0.19 | 1.4 ± 0.15 | 1.34 ± 0.12 | 1.3 ± 0.15 |
| Adam (const), 1e-4 | 2.15 ± 0.11 | 1.42 ± 0.15 | 1.12 ± 0.12 | 0.97 ± 0.1 | 0.81 ± 0.11 | 0.74 ± 0.1 | 0.72 ± 0.12 | 0.89 ± 0.13 |
| Adam (const), 1e-5 | 3.48 ± 0.06 | 3.13 ± 0.06 | 2.66 ± 0.07 | 1.96 ± 0.09 | 1.23 ± 0.1 | 0.96 ± 0.1 | 0.8 ± 0.1 | 0.72 ± 0.11 |

Table 20: Average and standard deviation evaluation loss for ViT experiment.

