# OpenReview forum: "Narrowing the Focus: Learned Optimizers for Pretrained Models"
_ICLR.cc/2025/Conference — Submitted to ICLR 2025_

### Official Review · Reviewer_wxWf · 2024-10-28

**Soundness:** 3
**Presentation:** 2
**Contribution:** 3
**Rating:** 5
**Confidence:** 4

**Summary:**

Typical model training requires an optimizer such as SGD, or ADAM and tune the hyper parameters such as learning rate. One alternative direction is to have meta-learning, where a general-purpose optimizer is trained from multiple tasks and then apply to the specific problem. This meta-learning approach such as VELO is computational expensive. This paper tries a different direction which is to learn a specialized learned optimizer namely L3RS to focus on improving the pre-trained model fine-tuning. Different from the VELO approach, the proposed L3RS uses smaller MLP networks for per-layer operations and relies on base optimizers. This design choice leads to two to three orders magnitude fewer parameters compared to the VELO. The proposed L3RS shows 50% speed up over VELO and 100% speedup over the best hand-designed optimizer on ImageNet benchmark.

**Strengths:**

1. L3RS is a unique learning-to-learn optimizer approach that focuses on the fine-tuning cases. This is different from the standard fine-tuning optimizer such as SGD or ADAM, and it is also very different from the general learning-to-learn optimizer such as VELO.
2. The proposed approach is evaluated on various experimental settings such as ImageNet and PLACES datasets. The proposed experimental setting seems to be typical for learning-to-learn related papers.
3. L3RS shows a significantly speed up than the VELO and baseline in the benchmarks.

**Weaknesses:**

1. Experiments are not comprehensive and convincing. The actual baseline used in the experiments is VELO. However, the field has many other famous approaches, such as Shampoo optimizers. Can the proposed approach show better results than Shampoo as well? I am also not sure about the generalization of the proposed approach. Specifically, how would one know that the layers trained with the proposed approach are not stuck in the local optimal? Can you provide any theoretical or experimental analysis? It seems to be odd to focus only on achieving accuracy. It would be better to see if the model is converged.
2. Resources. The proposed approach is still considerably expensive, and it might not be that attractive to the practitioner if it only helps to optimize the optimizer to adapt to the specific dataset with specific model fine-tuning. It would be great to justify the resources.

**Questions:**

1. How is the proposed approach perform against the other alternative such as Shampoo optimizer?

2. How is the proposed approach determine if the learned optimizer is useful to help the model to be trained to converged rather than reaching certain accuracy?

3. Can you give more back envelop estimation on the computational cost versus the benefits of the proposed approach bringing in in the actual useful applications?

---

> ### Author Response · Authors · 2024-11-26
>
> > “However, the field has many other famous approaches, such as Shampoo optimizers.”
>
> We appreciate the reviewer's suggestion. We chose VeLO as it is the SoTA learned optimizer at this time. Shampoo is a traditional optimizer, but is worthwhile to consider as a baseline. Due to resource constraints we chose a limited number of baselines. We compared to Adam since it is a popular, robust method which tends to perform well with little hyperparameter tuning.
>
> > “How is the proposed approach determine if the learned optimizer is useful to help the model to be trained to converged rather than reaching certain accuracy?”
>
> While model convergence is an important aspect for an optimizer, guaranteeing global convergence is a challenge for most optimization algorithms, including widely used methods like Adam (see e.g. https://arxiv.org/abs/1904.09237). Our method, like most other meta-optimization approaches, focuses on practical performance improvement rather than theoretical convergence guarantees.
>
> However, because of the interpretability of our method, we can demonstrate certain aspects of the final performance that hints at convergence of our model (see Fig.9 from the appendix). First, close to the end of training, the learning rate drops significantly, which aligns with the principle of gradually reducing the learning rate as the model approaches a minimum. Second, our learned algorithm switches from Adam to SGD, which helps with the convergence. This transition leverages the strengths of each optimizer: Adam for initial exploration in early stages, and SGD for fine-tuning and potential convergence to sharper minima in later stages. This strategy is consistent with established practices in deep learning (see e.g. https://arxiv.org/abs/1705.08292).
>
> These clarifications and additions will be incorporated into the final version of the manuscript.
>
> > “It seems to be odd to focus only on achieving accuracy. It would be better to see if the model is converged.”
>
> Accuracy is the preferred metric for the datasets we’ve chosen, but we have included corresponding figures and tables using loss as the metric in the appendix. This allows for a comprehensive evaluation of performance beyond just accuracy.
>
> > “Can you give more back envelop estimation on the computational cost versus the benefits of the proposed approach bringing in in the actual useful applications?”
>
> Table 1 provides the memory and compute overheads for our optimizer. While overall training a meta-optimizer is computationally more expensive, we argue that the benefits of our meta-training approach outweigh the costs in several practical scenarios, for example.
>
> 1. Fine-tuning large models often involves thousands of iterations with different datasets or tasks. Assuming a typical fine-tuning run requires $X$ compute hours and our meta-trained optimizer improves performance by $Y\%$, amortizing the meta-training cost ( $Z$ compute hours) becomes beneficial after approximately $Z / (X * Y/100)$ fine-tuning runs. Given the scale of the model deployments (e.g. for LLM), this threshold can be easily surpassed.
>
> 2. In domains where even small performance gains translate to substantial real-world impact (e.g., medical diagnosis, fraud detection), the investment in meta-training is justified. A $X\%$ improvement in accuracy, even at a higher initial computational cost, can lead to significant downstream benefits that outweigh the initial investment.
>
> 3. In cases where data or compute resources are limited, efficiently utilizing the available resources is critical. Our method, by achieving faster convergence and potentially better performance with fewer fine-tuning steps, directly addresses this constraint. For instance, if our method reduces the required training steps by $X\%$, this translates to a proportional reduction in time and cost, making fine-tuning feasible in scenarios otherwise impossible.

---

> > ### Comment · Reviewer_wxWf · 2024-11-27
> >
> > 1. It is understandable to skip Shampoo this time. However, if things are allowed, i feel that comparison would help to provide a good perspective on the proposed optimizer. I am not sure why the 2nd order optimizer is skipped, given that has done experiments on the same if not the similar dataset. See https://arxiv.org/pdf/1802.09568. The experiments were running on CIFAR 10/100 in Shampoo paper back in 2018. It seems that the 2024 GPU resources that could be running for the proposed optimizer on ImageNet can be easily reused to run the experiments on CIFAR 10/100?
> >
> > 2. Accuracy might makes sense in the selection application, it might be very misleading to have the practical meaning in a lot of application. e.g. autonomous vehicle application, it makes very little sense to have a model to reach one accuracy without looking into the other metrics such as confusion matrix, precision, recall, etc. I would like to learn more about what the motivation of only looking into accuracy.
> > 3. By back envelope estimation, i would like to encourage authors to list the estimation of the computational costs and other costs associated with applying the proposed optimizers in the real application. Yes, i agree with the arguments that it might worth paying the cost wit the proposed optimizer when we consider some circumstances. But I am also interested in learning the weakness of the proposed approach, and see if there is any rooms to be improved.

---

> > > ### Author Response · Authors · 2024-11-28
> > >
> > > > It is understandable to skip Shampoo this time.
> > >
> > > We appreciate the understanding, but would like to make it clear that we agree Shampoo is a very reasonable baseline. Time and engineering resources were the main constraints that prevented us from adding Shampoo as a baseline during this review process, but we agree it would be ideal to add and thank the reviewer for the suggestion.
> > >
> > > > Accuracy might makes sense in the selection application, it might be very misleading...
> > >
> > > We agree that accuracy would be a misleading metric in many other applications. One of the reasons we chose ImageNet as a dataset is that its a popular benchmark for image classification and there is a main interpretable metric, accuracy. Many researchers are familiar with ImageNet and what various accuracies mean with respect to the model performance. It is true that accuracy is not the full story, which is why we include loss as well. We are more than happy to discuss this further if this wasn't a satisfactory answer or if we misunderstood, and would like to understand what metrics you might prefer to see.
> > >
> > > > By back envelope estimation, i would like to encourage authors to list the estimation of the computational costs and other costs associated with applying the proposed optimizers in the real application.
> > >
> > > Is the suggestion here that we provide specific computational costs and resources required to practically apply this method? If so, the difficulty with doing so is that its extremely dependent on the application. Model size and inference/gradient cost are the largest factors in the actual cost of practical application of the method. The optimizer itself is extremely lightweight relative to many models used in practice today. Our intention in the previous response was to give some rules of thumb to determine when the method would be practical in application. Due to the cost of meta-training it certainly is not always worthwhile, but we believe the scenarios we described where it is worthwhile cover a large breadth of practical settings.
> > >
> > > We thank the reviewer for their time and feedback look forward to continuing this discussion.

---

### Official Review · Reviewer_qsSB · 2024-11-02

**Soundness:** 3
**Presentation:** 1
**Contribution:** 2
**Rating:** 5
**Confidence:** 4

**Summary:**

This work proposes a learning to learn method where learned optimizers are used to finetune pretrained models. The work focuses mainly on downstream tasks with the goal of focusing the meta learning process on narrower tasks. Experiments revolve around showing that the learned optimizer can use off-the-shelf optimizers to surpass both off-the-shelf and SotA learned optimizers.

**Strengths:**

The experiments on task distribution are comprehensive.
The section on preliminaries is comprehensive.
The ablation studies show the effectiveness of each design choice.

**Weaknesses:**

- The abstract is poorly written with wording that replaces necessary definitions. What is meant by “transforms the updates” (missing context), “various statistics” (can be replaced with an actual example) or “hand-designed” (didn’t understand what this could be in relation to)? In general, in terms of writing, the paper seems rushed and incomplete with inconsistencies in writing quality.

- L154: There is a mention of "a lot of parameters". A casual writing style isn't appropriate. Either the number of parameters should be mentioned or a relative figure should be given justifying the issues that the authors take with VELO. Especially since this paper quantitatively addresses this issue in Table 1 and a simple referral to that section would be enough.

- L153: "2-hidden layer, 4-hidden MLP" this part doesn't make sense. 4-hidden MLP?

- L155: The usage of the word "lean" is vague and uncommon as far as I'm aware. Did you mean flexible?

- L189: “is probably the closest”. Again, casual writing style is not appropriate.

- The structuring of paragraphs is odd. As an example, in the preliminaries section, some paragraphs are only a single sentence.

- One of the problems mentioned in the intro is about the short horizon and how far we need to move from the pretrained model to train it. But then in the next few paragraphs, the authors mention that they want to focus on finetuning pretrained models. Wouldn’t that make the problem of short horizons moot not only for this work but also for the previous works that supposedly had problems with long horizons?

- I didn’t find a comparison to ATMO Landro et al. (2021) in the paper especially since it is mentioned in the paper that ATMO is the closest to this work. Is the ablation at L476 a representation of ATMO?

- The experiments seem to focus on one structure and one variant of that structure. More structures are needed to show that the approach isn't limited to resnet only especially since more complicated mechanisms like attention don't exist in ResNet-34 making them unsuitable for a considerable number of current day problems.

- Section 5, meta-evaluation and baselines: why are those baselines chosen? there is no description of the reasoning as to what the point of comparison to those baselines is. This is referring to the case of cosine, cosine head, const and const head. If the point is to vary the learning rate scheduler, there are many other schedulers. What makes cosine scheduling so special to act as a baseline? Is it the popularity of cosine schedulers? To note, I'm asking for a description for the reasoning of choosing those baselines. The VELO models have descriptions giving reasoning as to why they were chosen for comparison but the other baselines don't.

- Since the work is focused on narrow datasets and finetuning scenarios, I’d expected to see more datasets with varying granularity to showcase the effectiveness of this method.

Nitpicks:
Tables should be changed to booktabs format for readability if the venue rules allow.
Figure 3 caption is hard to read.
L200, comma missing at the start of paragraph.

**Questions:**

Please refer to the weaknesses section.

---

> ### Author Response · Authors · 2024-11-26
>
> We appreciate the reviewer for the careful review and for pointing out many small errors and typos that we missed. We will certainly address and fix them as well as carefully re-review for any others missed.
>
> > “One of the problems mentioned in the intro is about the short horizon and how far we need to move from the pretrained model to train it.”
>
> Thank you for raising this important point. We understand the apparent contradiction between highlighting the short horizon problem and then focusing on fine-tuning. Allow us to clarify.
>
> While fine-tuning pretrained models does typically involve shorter training times compared to training from scratch, it does not inherently eliminate the short horizon bias. There are situations where we need to adapt to a slightly longer optimization trajectory than what the meta-optimizer was originally trained on. For example, we might encounter a new task or dataset that requires a few more fine-tuning steps to reach optimal performance. A meta-optimizer trained on very short horizons might fail to generalize effectively, even if the absolute fine-tuning time remains relatively short. In our work, we demonstrate that our proposed optimizer is robust to this problem. Specifically, we show that it can successfully generalize to longer fine-tuning horizons than those it was explicitly meta-trained on (e.g. see Figure 3 meta-generalization of L3RS curves from solid to dotted lines).
>
> > “I didn’t find a comparison to ATMO”. Is the ablation at L476 a representation of ATMO?
>
> Thanks for raising this. The ablation at L476 and "Global" in Table 2 are our closest comparisons to ATMO, as they reduce our method to a weighted combination across all layers, similar to ATMO's linear combination of SGD and Adam.
>
> However, even here our method is meta-learned, the linear weighting and learning rates are chosen dynamically, while ATMO uses a static hyperparameter. Thus, "Global" provides an upper bound on ATMO's performance. We'll clarify this in the manuscript.
>
> > “The experiments seem to focus on one structure and one variant of that structure”
>
> We acknowledge that evaluating on a wider range of architectures is an important consideration. We initially focused on ResNet-34 due to its widespread use and established performance as a benchmark in the field. Furthermore, we don’t expect any intra-model generalization, since our current focus is fine-tuning from a fixed checkpoint and we expect some degree of model-specific adaptation is expected and, in some cases, even beneficial. We will be sure to highlight this scope and the potential for future architectural exploration in the revised manuscript.
>
> > “To note, I'm asking for a description for the reasoning of choosing those baselines.”
>
> Our choices were guided by both standard practice and the need to isolate specific contributions. Adam serves as a robust and widely used optimizer benchmark. Cosine Schedule reflects the increasing popularity of cosine learning rate decay in fine-tuning for its smooth convergence. While other schedulers exist, the cosine schedule's effectiveness and prevalence made it a representative choice. Constant Learning Rate isolates the benefits of a schedule versus a single, fixed or learned global learning rate. 'Const head' adds a minimal learnable scaling in cases the budget is limited. We will clarify this reasoning in Section 5 of the revised manuscript.
>
> > “I’d expected to see more datasets with varying granularity”
>
> Thank you for raising this important point about the diversity of datasets used in our evaluation. We understand the desire to see L3RS applied to a wider range of datasets with varying granularities. Our current focus, however, is on demonstrating the effectiveness of L3RS on established, large-scale image classification benchmarks (ImageNet and PLACES). This allows us to conduct a more in-depth analysis (like ablation in studies in Section 6) and demonstrate the effectiveness of L3RS within this specific context. We will clarify this scope and the motivation for our dataset selection in the revised manuscript.

---

> ### Comment · Reviewer_qsSB · 2024-11-27
>
> Thank you for your detailed answers. I have a few questions and points to raise.
>
> > For example, we might encounter a new task or dataset that requires a few more fine-tuning steps to reach optimal performance.
>
> As mentioned by the reviewer Gurv, in longer horizons, the proposed approach only outperforms Adam in the early stages with smaller training steps and on longer horizons, Adam works fine. So based on your answer to that comment and mine, would that mean that the specific targeted problem here would be very short horizon + few-shot or continual learning?
>
> > We initially focused on ResNet-34 due to its widespread use and established performance as a benchmark in the field
>
> I would accept this answer in the light of the paper having deeper insights. However, the cited works that have been mentioned as being close to the submitted work seem to experiment with more than one structure. [a] has experimented with ResNet18, ResNet34, and BERT. [b] has experiments for GPT2, ResNet18, ResNet50. So I'd at least expect some potential results with one other structure on commonly used datasets but without any other structures, I wouldn't be comfortable changing my score.
>
> I'm happy with the answers to the rest of my initial questions. Although, my comments about the writing of the paper have not been addressed. I expected the manuscript to be updated with better writing by now.
>
> [a] Landro N, Gallo I, La Grassa R. Combining Optimization Methods Using an Adaptive Meta Optimizer. Algorithms. 2021; 14(6):186. https://doi.org/10.3390/a14060186
>
> [b] Almeida, D., Winter, C., Tang, J., & Zaremba, W. (2021). A Generalizable Approach to Learning Optimizers. ArXiv, abs/2106.00958.

---

> > ### Author Response · Authors · 2024-11-28
> >
> > We are happy to hear that this discussion has been beneficial and has answered some of your questions. We are also sincerely thankful for the suggestions made. We have updated our paper with experiments on ResNet-18 and Vision Transformers. By demonstrating the method across three architectures, we believe the paper is stronger.
> >
> > >  ...would that mean that the specific targeted problem here would be very short horizon + few-shot or continual learning?
> >
> > These certainly are settings where this method is useful. On-device model tuning is also a scenario where this method could prove practical. The optimizer could be meta-trained ahead of time using more computational resources than would be available on the device. Our method would be especially useful if a single learning rate schedule did not perform consistently across tasks. Our method may be able to dynamically adjust to the training dynamics online, whereas traditional optimizers would need to be tuned by retraining.
> >
> > We look forward to continuing this discussion and we hope that you would consider updating the score in light of the new experiments.

---

### Official Review · Reviewer_Gurv · 2024-11-03

**Soundness:** 2
**Presentation:** 3
**Contribution:** 2
**Rating:** 3
**Confidence:** 3

**Summary:**

This paper explores learned optimizers to refine pretrained models. The proposed method, L3RS, dynamically adjusts the weighting scaler for gradient directions derived from different optimizers like SGD and Adam in a layer-specific manner and tunes hyperparameters such as the learning rate during the meta-learning process. The results demonstrate that L3RS surpasses both traditional and other learned optimizers in terms of performance during the initial fine-tuning steps.

**Strengths:**

1) The paper studies the important challenge of automating hyperparameter tuning for optimizers, which is a critical aspect of training neural networks efficiently.
2) By integrating the update directions from multiple optimizers and learning the optimal combination, the proposed method introduces a promising and potentially more adaptable approach to optimizer design.
3) L3RS shows enhanced performance in the early stages of fine-tuning compared to baseline methods.

**Weaknesses:**

1) My main concern is the applicability of this approach. According to Figure 3, the proposed learned optimizer only outperforms Adam in the early stages with smaller training steps. For larger training steps, Adam achieves comparable performance to the learned optimizer. Given that the learned optimizer requires extra time and memory to learn additional parameters, using it may not be necessary. Furthermore, the total convergence time, including the meta-learning process, may be slower than Adam.
2) The evaluation is limited to training a ResNet-34 model on image classification tasks. Expanding the experiments to include diverse tasks such as language understanding or generation and different architectures like transformers would provide a more comprehensive evaluation.
3) The study does not consider several recent and more effective variants of adaptive learning rate methods, such as Adabelief [1]. Including these in the comparative analysis or integrating their strategies could enhance the performance of learned optimizers and better justify their adoption.

[1] Zhuang, Juntang, et al. "Adabelief optimizer: Adapting stepsizes by the belief in observed gradients." Advances in neural information processing systems 33 (2020): 18795-18806.

**Questions:**

1)	Given that the final performance of L3RS is comparable to Adam and considering the extra computational costs, what are the specific scenarios or benefits that justify the use of learned optimizers?
2)	Experiments on additional tasks and models are necessary.
3)	A more in-depth discussion of advanced adaptive learning methods, such as Adabelief, is helpful.

---

> ### Author Response · Authors · 2024-11-26
>
> > “the proposed learned optimizer only outperforms Adam in the early stages with smaller training steps” … “Given that the final performance of L3RS is comparable to Adam and considering the extra computational costs, what are the specific scenarios or benefits that justify the use of learned optimizers?”
>
> While L3RS's final performance is comparable to Adam in long-horizon training, its key benefit lies in rapid adaptation. This makes L3RS particularly valuable in scenarios like few-shot learning, transfer learning, or when models require frequent and efficient fine-tuning on new data. The initial meta-training cost is amortized over numerous adaptation tasks, leading to overall computational savings in dynamic environments.
>
> > “Experiments on additional tasks and models are necessary.“
>
> We agree that evaluating our method on a broader range of tasks and models would be beneficial. Our choice of ImageNet and PLACES was driven by their recognition as challenging, large-scale benchmarks that effectively highlight the strengths of our method in a complex visual domain. Due to resource constraints, we focused on these core vision tasks for this study, but we intend to explore these avenues in future work.
>
> > “A more in-depth discussion of advanced adaptive learning methods, such as Adabelief, is helpful.”
>
> We appreciate the reviewer's suggestion to include a discussion of AdaBelief. Our choice of Adam was motivated by its status as a widely adopted and effective adaptive optimizer, providing a strong and relevant baseline for our method. While a comprehensive comparison with all advanced adaptive methods is computationally intensive, we agree that including AdaBelief is valuable. We will perform experiments with AdaBelief and provide a detailed analysis of the results in the appendix, including a discussion of its performance relative to Adam and our proposed method.

---

> > ### Comment · Reviewer_Gurv · 2024-12-03
> > **Post-rebuttal Comment**
> >
> > Thank you to the authors for their response. However, the majority of my concerns remain unresolved. The applicability of the proposed approach is still unclear. While the authors claim it may be useful in scenarios such as few-shot learning, there is insufficient evidence or experimental support to substantiate this claim. As a result, I maintain my original score.

---

### Official Review · Reviewer_76HK · 2024-11-07

**Soundness:** 3
**Presentation:** 2
**Contribution:** 2
**Rating:** 5
**Confidence:** 4

**Summary:**

This work introduces L3RS, a learned learning rate scheduler that ensembles updates from base optimizers like Adam and SGD.

L3RS is parameterized as a shared MLP and layer-wise trainable embeddings. For each layer, L3RS takes its embedding and training status statistics (training time, exponential moving average of loss/gradient norm/weight norm) as input, and predicts coefficients of base optimizer updates. L3RS is trained with natural evolution strategies. In various settings with ImageNet and PLACES dataset, L3RS shows higher classification accuracy and training efficiency on ResNet34 finetuning.

**Strengths:**

1. The presentation is clear and easy to understand; The experiments are comprehensive and well-explained.
2. The proposed method is novel and useful: it eliminates the need for learning rate schedule tuning.
3. The proposed method demonstrates faster convergence and improved performance.

**Weaknesses:**

1. L3RS is not generalized and needs to be trained independently for each model architecture, which limits its application. From ImageNet <-> PLACES experiments it seems a trained L3RS can generalize to other data distributions, and it would be more beneficial to include more benchmarks to showcase that.
2. Experiments are only conducted on ResNet34, while including evaluation on other architectures and more diverse benchmarks (e.g. [VeLO](https://arxiv.org/abs/2211.09760)) would make the results more convincing.
3. Finetuning for only 1000 steps may not well represent the practical finetuning setting.
4. Lack of a deeper analysis on how SGD/Adam update directions are favored by different layers/learning stages.
5. For practical use of L3RS, there needs to be a L3RS training recipe generalized across different model architectures, or pretrained L3RS checkpoints tuned for popular architectures.

**Questions:**

See weaknesses

---

> ### Author Response · Authors · 2024-11-26
>
> > “L3RS is not generalized and needs to be trained independently for each model architecture”
>
> We agree that exploring the broader generalization of L3RS across diverse architectures is a valuable direction for future work. However, it's important to note that our current focus is on the specific and challenging problem of fine-tuning from a fixed checkpoint. In this domain, some level of model-specific adaptation is often necessary due to the inherent differences between pre-trained models and downstream tasks. While L3RS is designed to be adaptable, our primary contribution lies in its effectiveness within this specific fine-tuning setting. We plan to investigate broader architecture generalization in future research.
>
> > “From ImageNet <-> PLACES experiments it seems a trained L3RS can generalize to other data distributions, and it would be more beneficial to include more benchmarks to showcase that.”
>
> We understand that more evaluations are always preferred. We chose ImageNet and PLACES as the evaluation datasets due to their complexity and we believe that the current experiments reasonably show the usefulness of the method. The ideal usage of the method is to meta-train the optimizer on the data-distribution it will be used on. This generalization outside of the training distribution is not always expected with meta-learning methods and showcases the robustness of the method.
>
> > “Experiments are only conducted on ResNet34, while including evaluation on other architectures and more diverse benchmarks (e.g. VeLO) would make the results more convincing.”
>
> We agree that evaluations of other architectures would be ideal, but have left that to future work. The VeLO paper did a good job of this, but VeLO is a general optimizer and needed to show meta-test time generalization to other architectures. The Velodrome dataset they created is also not applicable here because that dataset samples model architectures.
>
> > “Finetuning for only 1000 steps may not well represent the practical finetuning setting.“
>
> We agree that the optimal number of fine-tuning steps can vary depending on the specific application. However, our experiments demonstrate that L3RS is highly effective even with relatively short fine-tuning durations, achieving strong results with up to 2500 steps. Moreover, our results show that L3RS achieves significant performance gains even within the 500-2500 step range, which is outside of the meta-training distribution, indicating its efficiency and effectiveness in practical fine-tuning scenarios.
>
> > “Lack of a deeper analysis on how SGD/Adam update directions are favored by different layers/learning stages.”
>
> There are some patterns that we see with how the SGD/Adam directions are chosen throughout training. Figures 4, 8, and 9 show averages through training. In particular, Figure 8 shows the average parameter movement across all layers, reveals several distinct phases: an initial warm-up period with an increasing learning rate, a period of relatively constant learning rate while transitioning from ADAM to SGD, a phase of rapid learning rate decay, and a final convergence to the SGD direction over the last ∼10 steps. While the precise
> interpretation of these parameter dynamics is challenging, the L3RS parameters are considerably more interpretable than those of most black-box learned optimizers.
>
> > “For practical use of L3RS, there needs to be a L3RS training recipe generalized across different model architectures, or pretrained L3RS checkpoints tuned for popular architectures.“
>
> Thank you for raising this important point about the practical application of L3RS. We agree that generalizability and ease of use are key for adoption. Regarding hyperparameter selection, we designed our meta-training procedure with broad applicability in mind, and we believe the chosen hyperparameters will be effective across a range of scenarios. To clarify this, we will expand the paper with a detailed discussion of the rationale behind our hyperparameter choices and provide guidance on potential adjustments for specific use cases.
>
> We also strongly agree with the suggestion of providing pretrained L3RS checkpoints. Our vision for L3RS is indeed to offer a collection of specialized optimizers, tailored for different architectures and tasks, rather than a single monolithic model, like VeLO. We plan to release pretrained checkpoints for several popular architectures to facilitate practical use. Finally, we find the suggestion of investigating few-shot fine-tuning very compelling. This approach could enable rapid adaptation of L3RS to new models and data distributions, and we will explore this direction in future work.

---

### Author Response · Authors · 2024-11-28

We appreciate the reviewers taking the time to provide such detailed and helpful feedback. We have revised the paper to address your comments. We feel these changes have made our paper stronger and are truly thankful for the suggestions.

Specifically, we:

**Added experiments on ResNet-18 and Vision Transformer models.** A shared theme in the feedback was the concern about the limited range of model architectures in the initial submission. We hope these new results demonstrate the effectiveness of our approach beyond just ResNet-34.

**Included AdaBelief as a baseline.** We agree that AdaBelief is a relevant and important baseline, and we have now incorporated it into our experiments.

**Added a more thorough analysis of the SGD/Adam direction choices made by L3RS.** We believe that interpretability is a valuable quality for a learned optimizer. While we had some light analysis into the behavior of the optimizer, we agree that a more thorough analysis was needed. We hope that the new analysis is more clear and insightful.

We hope you agree and will consider raising your score to reflect these improvements. We are confident that our work makes a valuable contribution to the field of learned optimizers and would be a good fit for ICLR 2025.

Thank you again for your constructive feedback. We are grateful for the opportunity to improve our work based on your suggestions.

---

### Meta-Review · Area_Chair_GKJo · 2024-12-23

**Metareview:**

The submission introduces L3RS (Learned Layer-wise Learning Rate Scheduler), an optimizer that learns to combine the updates provided by hand-designed optimizers such as SGD and Adam, as well as heuristics such as multi-scale EMA.
After the initial round of reviews, this received scores of 5, 3, 5, 5.
All reviewers raised the concern that the method was not evaluated on a sufficient number of datasets, architectures, and tasks.

Most importantly, during the rebuttal, the authors uploaded a revision that de-anonymized the submission and revealed their names and affiliations. This was noticed and flagged by the reviewers and AC. This is in violation of the reviewing policy and is sufficient grounds for desk rejection.

**Additional Comments On Reviewer Discussion:**

All reviewers raised the concern that the method was not evaluated on a sufficient number of datasets, architectures, and tasks. Further, they pointed out that the model's loss using this new optimizer was comparable to Adam, given sufficient number of updates. Even after the rebuttal by the authors, the reviewers remained unconvinced.

---

### Decision · Program_Chairs · 2025-01-22

Reject